# Structural basis of antibody inhibition and chemokine activation of the human CC chemokine receptor 8

Dawei Sun[1], Yonglian Sun[2], Eric Janezic [3], Tricia Zhou [3], Matthew Johnson[1], Caleigh Azumaya[1], Sigrid Noreng[1,6], Cecilia Chiu[2], Akiko Seki[4,7], Teresita L. Arenzana[4,8], John M. Nicoludis [1], Yongchang Shi[3], Baomei Wang[4], Hoangdung Ho[1], Prajakta Joshi [5], Christine Tam[5], Jian Payandeh [1,9], Laëtitia Comps-Agrar [3], Jianyong Wang[3], Sascha Rutz [4] ✉, James T. Koerber [2] ✉ & Matthieu Masureel [1] ✉

The C-C motif chemokine receptor 8 (CCR8) is a class A G-protein coupled receptor that has emerged as a promising therapeutic target in cancer. Targeting CCR8 with an antibody has appeared to be an attractive therapeutic approach, but the molecular basis for chemokine-mediated activation and antibody-mediated inhibition of CCR8 are not fully elucidated. Here, we obtain an antagonist antibody against human CCR8 and determine structures of CCR8 in complex with either the antibody or the endogenous agonist ligand CCL1. Our studies reveal characteristic antibody features allowing recognition of the CCR8 extracellular loops and CCL1-CCR8 interaction modes that are distinct from other chemokine receptor - ligand pairs. Informed by these structural insights, we demonstrate that CCL1 follows a two-step, two-site binding sequence to CCR8 and that antibody-mediated inhibition of CCL1 signaling can occur by preventing the second binding event. Together, our results provide a detailed structural and mechanistic framework of CCR8 activation and inhibition that expands our molecular understanding of chemokine - receptor interactions and offers insight into the development of therapeutic antibodies targeting chemokine GPCRs.

The C-C motif chemokine receptor 8 (CCR8) is a class A G protein-coupled receptor (GPCR) that is highly enriched and selectively expressed on intratumoral regulatory T (Treg) cells, which act as suppressors of anti-tumor effector T cell responses[1–3]. Patients with higher levels of Treg cells exhibit poorer clinical outcomes and prognoses in several cancers[4,5]. As such, it has been hypothesized that selective depletion of intratumoral Treg cells could reinvigorate anti-tumor immune responses and improve responses to cancer immunotherapy. Recent preclinical mouse experiments have demonstrated that depletion of mouse Treg cells using an anti-murine CCR8 antibody can result in strong anti-tumor responses[3,6,7].

[1]Department of Structural Biology, Genentech Inc., South San Francisco, CA 94080, USA. [2]Department of Antibody Engineering, Genentech Inc., South San Francisco, CA 94080, USA. [3]Department of Biochemical and Cellular Pharmacology, Genentech Inc., South San Francisco, CA 94080, USA. [4]Department of Cancer Immunology, Genentech Inc., South San Francisco, CA 94080, USA. [5]Department of Biomolecular Resources, Genentech Inc., South San Francisco, CA 94080, USA. [6]Present address: Septerna Inc., South San Francisco, CA 94080, USA. [7]Present address: Tune Therapeutics, Durham, NC 27701, USA. [8]Present address: HIBio, South San Francisco, CA 94080, USA. [9]Present address: Exelixis Inc., Alameda, CA 94502, USA. ✉e-mail: rutz.sascha@gene.com; koerber.james@gene.com; masureel.matthieu@gene.com

The high potency, long half-life, and exquisite selectivity of monoclonal antibodies (mAbs) have made them successful therapeutics against many target classes. However, while approximately one third of approved drugs target GPCRs[8], only two GPCR-targeting mAbs have been approved so far[9,10], highlighting the challenges in generating and developing therapeutic antibodies against GPCRs. While both of these approved mAbs bind to the unstructured N-terminus of their respective receptors, likely in a manner similar to traditional antibody-peptide complexes, the recognition of conformational epitopes composed of the highly dynamic extracellular loops (ECLs) is likely required for the successful development of antibody antagonists or agonists against other GPCRs. New antigen formats and microfluidic technologies have improved GPCR antibody discovery and begun to reveal features underlying mAb binding to the ECLs. Features such as a long complementarity-determining region (CDR) H3 or the convex paratope of single domain antibodies appear to be important, and our structural understanding of how these contribute to GPCR recognition and how antibodies can modulate ligand binding has expanded in recent years[11–19].

CCR8 is part of the ten-member C-C motif chemokine receptor (CCR) subfamily of chemokine receptors[20]. While CCR8 is known to be activated by the endogenous C-C motif chemokine ligand 1 (CCL1) and coupled to the inhibitory signaling protein Gi[21], its molecular structure and activation mechanism remain unclear. Among members of the CCR subfamily, only CCR2 and CCR5 have been structurally characterized in both inactive and active states[22–27], while CCR7 and CCR9 inactive-state structures and CCR1 and CCR6 active-state structures are also available[28–31]. However, so far no structure of CCR8 has been reported, limiting our understanding of its specific activation mechanism by CCL1 and of its targeting and inhibition by antibodies. We therefore hypothesized that a better understanding of the structure of CCR8 may provide new insights into its function and potentially facilitate the generation of future therapeutics aimed at this emerging target.

Here, we generate mAb1, an antagonist antibody that binds the extracellular region of human CCR8. We then determine the structures of the fragment–antigen binding region of mAb1 (Fab1) bound to CCR8 and of the CCL1-CCR8-Gi signaling complex to provide key molecular insights into mAb1-mediated inhibition and CCL1-mediated activation of CCR8, thereby expanding our understanding of this pharmaceutical target. Furthermore, we provide cell-based binding results that support a two-step, two-site binding model of CCL1 to CCR8 and inform on the mechanism of mAb1-mediated inhibition of CCL1 signaling through CCR8.

## Results

### Generation and characterization of anti-CCR8 antibody mAb1

To study antibody-based inhibition of human CCR8, we generated the anti-CCR8 antibody mAb1 and characterized its binding and function. MAb1 detects the native conformation of human CCR8, as verified by its ability to bind to a subset of Treg cells present within human peripheral blood mononuclear cells (PBMCs) or dissociated tumor cells (DTCs) to a similar extent as a commercially available antibody (Supplementary Figs. 1, 2a). Additional cell surface binding experiments in transiently transfected HEK293 cells confirmed that mAb1 binds selectively to CCR8 but not to a panel of other chemokine receptors that also possess tyrosine sulfation sites, which are critical for CCR8 activity[32,33] (Supplementary Fig. 2b).

Scatchard analysis revealed that radiolabeled mAb1 bound to CHO cells stably expressing human CCR8 (CHO.hCCR8) with an affinity of 28.4 pM (Supplementary Fig. 2c). To map the mAb1 extracellular epitope on CCR8, we generated CCR8 chimeras, replacing the N-terminus or ECL1, 2, or 3 with the corresponding sequences from human CCR5, which we chose within the C-C subfamily based on its high sequence identity but fairly distal phylogenetic clustering to

CCR8. Flow cytometry analysis of mAb1 binding to HEK293 cells transiently transfected with each chimeric construct revealed that the ECL1 and ECL2 of CCR8, but not its N-terminus or ECL3, are required for mAb1 binding (Supplementary Fig. 2d).

Since mAb1 bound the ECL2 of CCR8, which is essential for chemokine binding[34], we wondered whether mAb1 could modulate CCR8 function. We thus performed a cell-based signaling assay to directly test whether mAb1 can antagonize CCL1-mediated CCR8 signaling. As expected, addition of CCL1 in the assay led to a dose-dependent increase in intracellular calcium, whereas addition of mAb1 alone did not change calcium levels, indicating a lack of mAb1 agonistic activity (Fig. 1a). Addition of mAb1 to a sub-maximal (EC80) concentration of CCL1, led to a dose-dependent decrease in calcium levels, indicating mAb1 antagonist activity, with an apparent IC50 of 57.9 nM (Fig. 1b). Addition of mAb1 to Jurkat cells stably expressing CCR8 led to a dose-dependent inhibition of CCL1-induced ERK phosphorylation, indicating that mAb1 blocked CCL1-mediated activation of the MAPK pathway (Fig. 1c and Supplementary Fig. 2e). Taken together, our results confirm that mAb1 binds selectively to a native epitope consisting of ECL1 and 2 on human CCR8 and thereby inhibits CCL1-induced agonism of CCR8.

### Overall structures of the Fab1-CCR8 complex and the CCL1-CCR8-Gi complex

We formed a stable and monodisperse complex between full-length human CCR8 and Fab1 and obtained a three-dimensional reconstruction by single-particle cryogenic electron microscopy (cryo-EM) with a nominal global resolution of 3.1 Å (Fig. 1d, Supplementary Figs. 3, 5a, c and Supplementary Table 1). The overall complex resembles a table lamp with shade, with Fab1 sitting atop the extracellular region of CCR8, which adopts the canonical class A GPCR fold characterized by a seven transmembrane helical bundle and limited extracellular regions (Fig. 1d).

Initial attempts to assemble a stable CCL1-CCR8-Gi complex using individually purified subunits proved unsuccessful. We therefore designed a CCL1-CCR8 fusion construct with an engineered disulfide bridge, similar to strategies previously used for the structural elucidation of CCR5 and CXCR4 complexes[27,35,36] (see Methods). Addition of Gi heterotrimer and scFv16[37] to the CCL1-CCR8 fusion combined with apyrase treatment yielded a stable, nucleotide-free complex which allowed us to obtain a three-dimensional reconstruction using single-particle cryo-EM with a nominal global resolution of 2.9 Å (Fig. 1e, Supplementary Figs. 4, 5b, d, e and Supplementary Table 1).

Comparison of both CCR8 structures indicate that in the Fab1-bound structure, the ECLs 1-2 and the intracellular loops (ICLs) 1-2 are well resolved, while the N-terminus, ECL3 and ICL3 are disordered (Supplementary Fig. 6a). In contrast, all regions of the receptor, with the exception of the first 19 N-terminal residues, are well-resolved in the CCL1-bound structure (Supplementary Fig. 6b). The receptor-proximal regions of CCL1, including its N-terminus, are well-resolved (Supplementary Fig. 5d), whereas CCL1 regions most distal to the receptor are not, suggesting a higher degree of conformational variability in these regions. The disulfide bridge between the N-terminus and the ECL3 at the end of transmembrane helix (TM) 7, critical for CCL1 signaling activity and conserved in chemokine receptors[34], is observed in the CCL1-CCR8-Gi structure but absent in our Fab1-CCR8 structure, likely due to disulfide bond instability during protein purification in the absence of chemokine[22,28] (Supplementary Fig. 6a, b). The disulfide bridge between TM3 and ECL2, conserved in class A GPCRs and essential for CCL1 binding[34], is present in both structures (Supplementary Fig. 6a, b). In both CCR8 structures, the N-terminal half of ECL2, termed ECL2a, adopts a β-hairpin fold which is commonly observed in chemokine receptors (Supplementary Fig. 7a), in a conformation that leaves the ligand-binding pocket of the receptor accessible[38] (Fig. 1d, e).

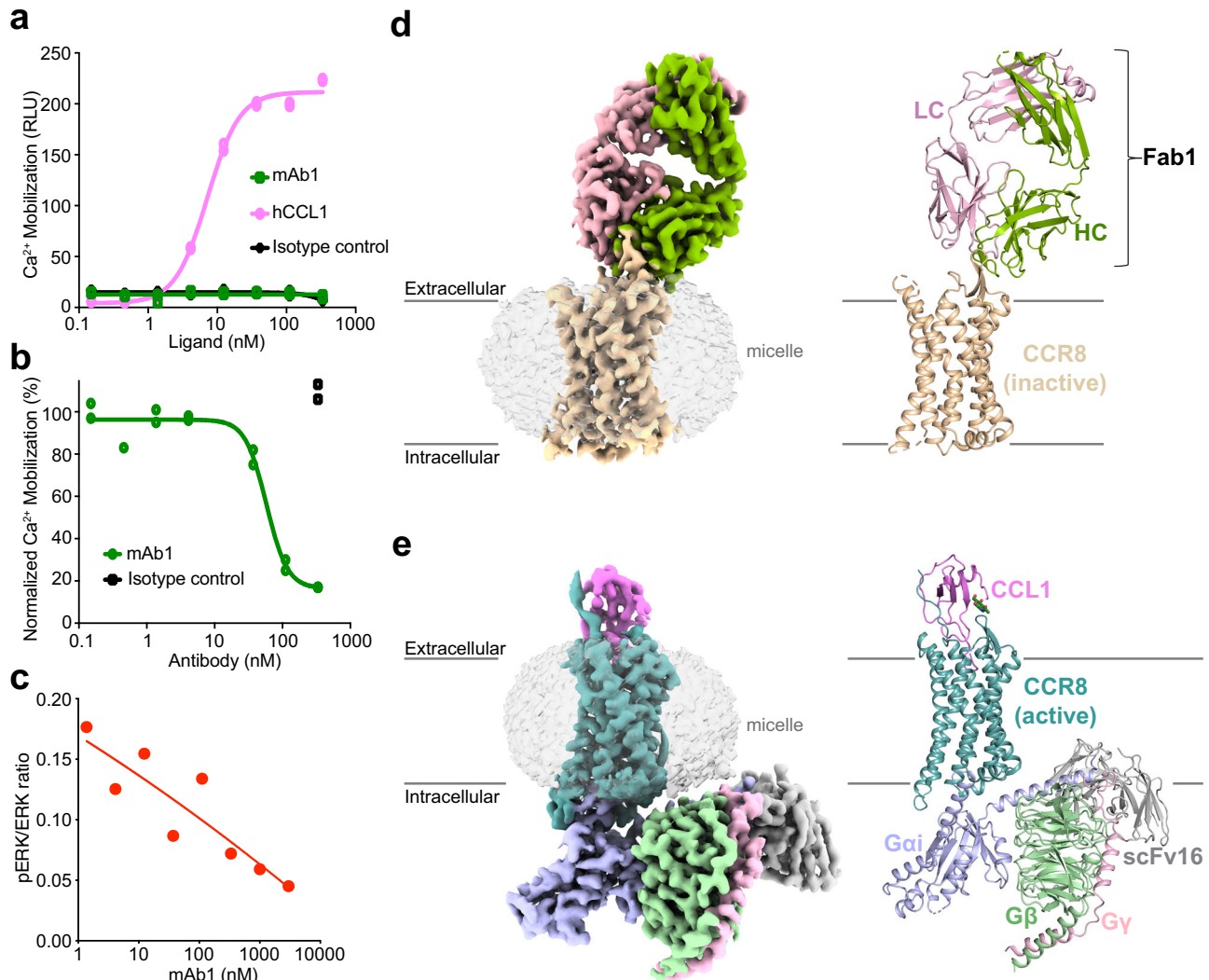

**Fig. 1 | Functional and structural characterization of mAb1 and CCL1 binding to CCR8.** Analysis of mAb1 and CCL1 binding on CCR8 activation (**a**) and of mAb1 competition binding on CCL1-induced CCR8 activation (**b**) by $Ca^{2+}$ influx cellular assay. Experiments were performed in duplicate, with error bars showing standard deviation. **c** Analysis of mAb1-mediated inhibition of CCL1-induced ERK phosphorylation, as analyzed by total ERK and phospho-ERK western blot as a function of mAb1 concentration. Data are representative of two independent experiments. **d** Cryo-EM map and model of the Fab1-CCR8 complex. **e** Cryo-EM map and model of CCL1-CCR8-$G_{i1}$-scFv16. Protein subunits in d and e are colored as follows: CCR8 inactive state, wheat; light chain (LC) of Fab1, pink; heavy chain (HC) of Fab1, green; CCR8 active state, teal; CCL1, magenta; $G\alpha_{i1}$, blue; $G\beta_1$, green; $G\gamma_2$, pink; scFv16, gray. Source data are provided as a Source Data file.

## Interactions between Fab1 and CCR8

Fab1 forms an extensive, mostly electrostatic, interaction interface with the ECLs 1 and 2 of CCR8 that is mainly mediated by its CDRH3, with additional contributions from CDRH1, CDRH2, CDRL1, and CDRL3 (~650 Å2, Fig. 2a). Close inspection of the interface reveals three key interaction regions. The first and main interaction interface is mediated by a continuous stretch of six residues in CDRH3 which forms numerous polar interactions with residues in the first β-strand and the β-turn of the ECL2a β-hairpin (Fig. 2c). These interactions create an antiparallel β-strand pairing, similar to what was observed in the structure of human 5-hydroxytryptamine 2B (5-$HT_{2B}$) receptor bound to an extracellular antibody[11], unlike any other structures of class A GPCRs in complex with a Fab engaging the receptor extracellular loops (Supplementary Fig. 7b). The Fab1-CCR8 interaction interface is further stabilized by additional polar interactions between CDRH3, CDRL1, CDRL3, and the ECL2b region at a second interface (Fig. 2b). Finally, a third interface involves Fab1 CDRH1 and 2 interacting with CCR8 ECL1 and the ECL1-facing side of the ECL2 β-hairpin through both polar and hydrophobic contacts (Fig. 2d).

Sequence conservation within ECL2 is low across the CCR family, rationalizing why mAb1 is highly specific and selective for binding CCR8 over other C-C chemokine receptors (Supplementary Fig. 7c). Notably, Fab1 does not interact with the CCR8 N-terminus or ECL3, which are not resolved in our structure, in agreement with our flow cytometry experiments (Supplementary Fig. 2d).

## CCL1−CCR8 interactions

The CCL1 ligand, glycosylated at N52 as expected[39], is intricately engaged with CCR8 through an interface formed by contributions from the receptor N-terminus and all three ECLs (Fig. 3a). This extensive interaction interface, mediated by multiple key polar interactions, can be grouped into previously defined chemokine recognition sites (CRS) 1, 1.5 and 2[26, 27,30,40] and a unique binding site within CRS2 that appears critical to CCL1-CCR8 binding, termed CRS2.5 (Fig. 3b−d).

The CRS1 is formed by a polar groove on the chemokine between the N-loop and the 40 s loop that engages the receptor N-terminal region preceding the conserved disulfide bridge C25$^{Nter}$−C272$^{7.25}$ (superscripts indicate Ballesteros-Weinstein numbering[41]), resolved

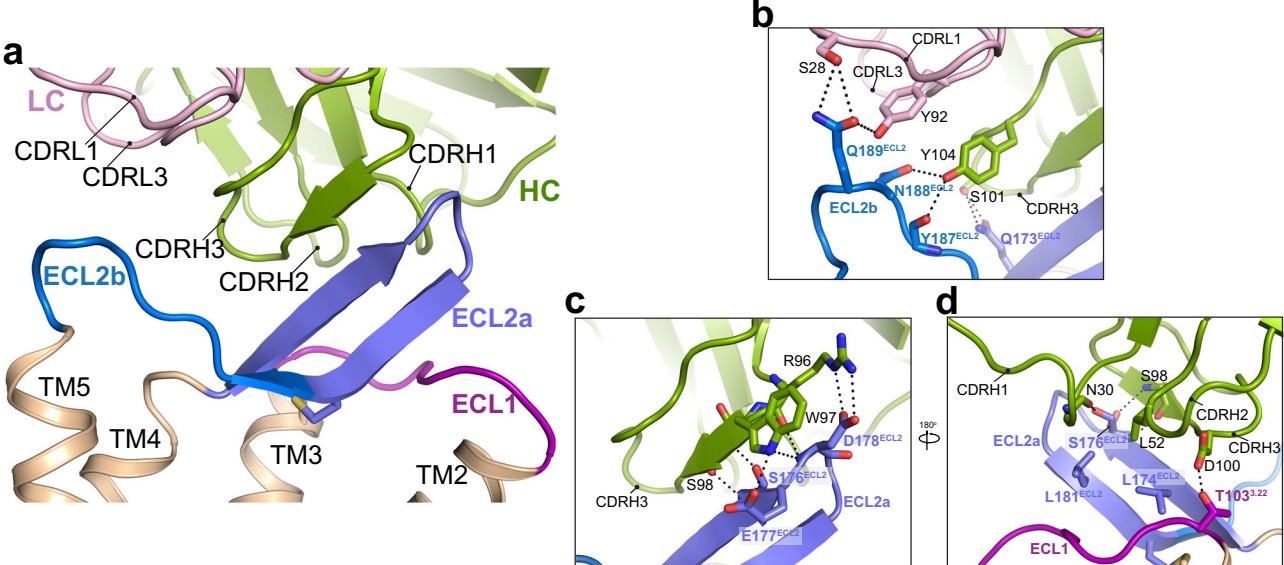

**Fig. 2 | Molecular interactions between Fab1 and CCR8. a** Overview of the Fab1 paratope and the CCR8 epitope. Close-up view of the interactions between CCR8 and Fab1, with major interactions formed between ECL2a of CCR8 and CDRH2,3 of Fab1 (**b**) and between ECL2b of CCR8 and CDRL1,3 of Fab (**c**, **d**). The ECL1, ECL2a, and ECL2b of CCR8 are colored plum, violet and blue respectively. Side chains forming interactions are shown as sticks and polar interactions are displayed as dashed black lines.

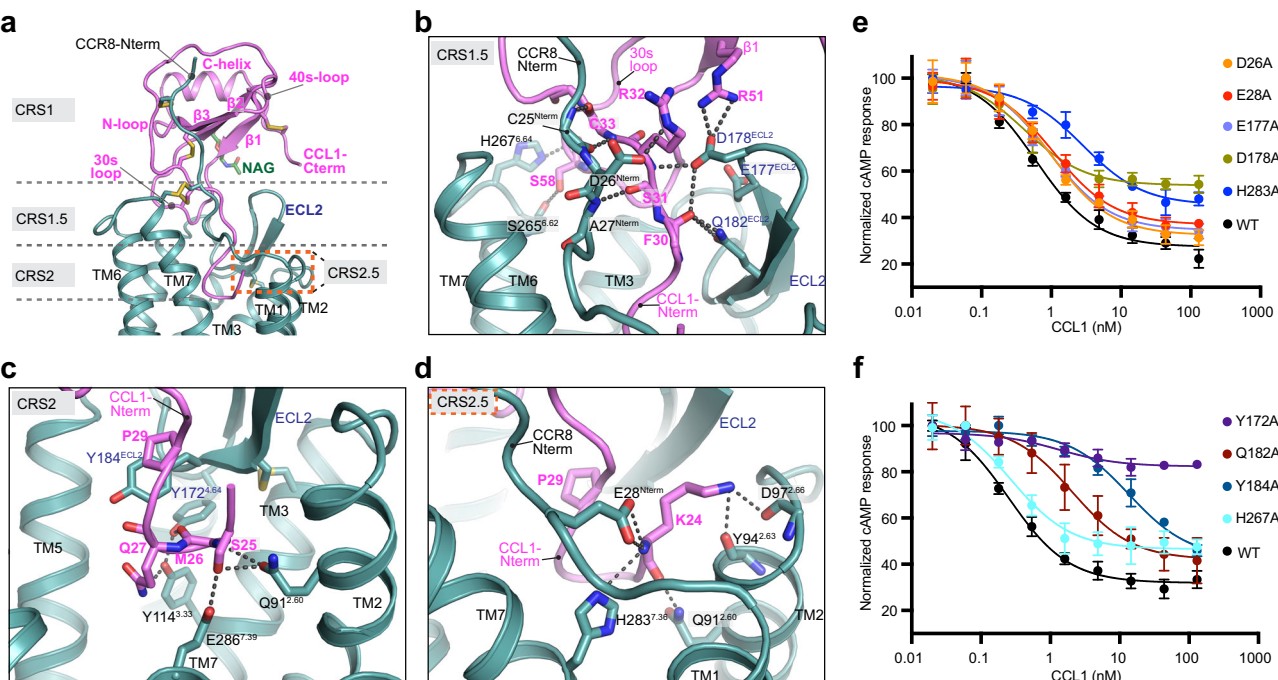

**Fig. 3 | CCL1 - CCR8 chemokine recognition sites. a** Overview of CCL1-CCR8 engagement, with CRS1, CRS1.5 and CRS2 delineated by dashed gray lines and the unique CRS2.5 region outlined by dashed orange lines. Close-up view and detailed interactions between CCR8 and the CCL1 N-terminus at CRS1.5 (**b**), CRS2 (**c**) and CRS2.5 (**d**). The CCL1 and its residues are colored in magenta, the glycan at CCL1 residue N52 shown in green and CCR8 colored teal. Disulfide bonds and key interacting residues are shown as sticks, with polar interactions shown as black dashed lines. Important CCR8 residues are labeled in black, except for ECL2 residues, which are labeled in blue. **e**–**f** Normalized cAMP levels measured on HEK293T cells transiently expressing WT or mutated CCR8 in the presence of 1 μM Forskolin and increasing concentrations of CCL1. Data are representative of two independent experiments performed in triplicates (mean ± S.D). Best-fit IC50 and Imax values along with relative receptor expression levels are reported in Supplementary Table 2. Source data are provided as a Source Data file.

from residue I20 onwards. While the first 12 residues of CCR8 are dispensable for CCL1 signaling activity[42], sulfation of the tyrosine cluster at residues 15–17 in human CCR8 is critical for CCL1 binding[32]. Stabilization of the ligand-receptor interactions through direct fusion and insertion of a disulfide bridge likely bypasses the required presence of sulfated tyrosine residues for receptor engagement. The CRS1.5, centered between the conserved P24-C25 motif on the CCR8 N-terminus and the β-turn region of ECL2 engage residues proximal to the CC motif on the chemokine N-terminus, with strong sidechain and backbone interactions (Fig. 3b).

The canonical CRS2 site, which corresponds to the distal CCL1 N-terminus engaging the receptor core, is stabilized by an extensive interaction network that involves CCR8 TMs 2, 3 and 7 and every CCL1 residue in this region (Fig. 3c). Notably, the CCL1 N-terminus folds back up away from the receptor core toward the extracellular regions, a feature that has not been observed in any other published chemokine–receptor structure so far but still affords a similar depth of receptor engagement (Supplementary Fig. 8), leading us to define an additional chemokine recognition site, termed CRS2.5 (Fig. 3d). In this CRS2.5 interface, the most N-terminal CCL1 residue K24 forms backbone–side chain interactions with CCR8 residues E28[1.25] and H283[7.36] on the N-terminus and TM7, respectively, and side chain–backbone interactions with CCR8 residues Y94[2.63] and D97[2.66] on TM2 and ECL1, respectively. Further completing these interactions, we also observe a minor interface where the CCL1 30 s loop engages CCR8 ECLs 2 and 3.

To further assess the role of receptor residues likely involved in ligand binding based on our structure, we performed functional studies on a representative set of CCR8 mutants (Fig. 3e, f; Supplementary Table 2). We observe that alanine mutants of CCR8 residues involved in CCL1 binding display varying degrees of functional impairment in a CCL1-induced cAMP inhibition assay, with mutations Y172A[4.64], D178A[ECL2], Q182A[ECL2], Y184A[ECL2], and H283A[7.36] having the strongest effect within the mutants we tested.

Molecular Dynamics simulations are frequently used to validate the chemokine binding pose and to rationalize the conformational heterogeneity for certain chemokine regions observed in experimental structures[26,27,30,31]. Thus, we performed Gaussian-accelerated Molecular Dynamics (GaMD) simulations[43] on the CCL1-CCR8-Gi complex embedded in a lipid bilayer (Supplementary Table 3). This unbiased simulation procedure uses enhanced sampling methods to sample conformational space more quickly, and confirmed a persistent interaction between the chemokine and the receptor, particularly at the CCL1 N-terminus, while also revealing a 20 degree heterogeneity in the relative chemokine · receptor orientation (Supplementary Fig. 9a–d). This conformational heterogeneity may explain the lower resolution of CCL1 in parts of our cryo-EM map, as previously observed in other chemokine-receptor structures[26,30]. Importantly, while removal of the engineered disulfide in our simulations allowed the CCR8 N-terminus to move away from CCL1, it did not alter the conformational heterogeneity of CCL1, suggesting that the engineered disulfide does not affect how CCL1 engages the receptor core.

## CCL1-induced CCR8 activation mechanism

Comparison of our inactive- and active-state CCR8 structures reveals how chemokine binding rearranges the ECL2 and moves the extracellular portions of TMs 1, 2 and 5 inwards to enable the stable insertion of CCL1's nine most N-terminal residues into the receptor core (Fig. 4a–c). Within the receptor core, we observe that residues Q91[2.60], Y114[3.33], Y172[4.64], and E286[7.39] rearrange due to their direct interactions with the CCL1 N-terminus and that residues Y113[3.32], M202[5.42], and F254[6.51], which do not interact with the ligand but are located directly below the most deeply penetrating CCL1 residues, also rearrange (Fig. 4d).

Inspection of residues that are part of the canonical GPCR microswitches[44, 45] reveals sidechain rearrangements underlying CCR8 activation (Fig. 4e). Notably, in the active-state structure we observe a methionine-aromatic interaction between M121[3.40], part of the PIF activation motif corresponding to residues P210[5.50]M121[3.40]S247[6.44] in CCR8, and the key toggle-switch residue W251[6.48]. In addition, we observe that rotation of TM6 in the active-state structure breaks the interaction between S247[6.44] and NPxxY motif residue N296[7.49], allowing local rearrangement of NPxxY motif residues P297[7.50] and Y300[7.53]. Accompanying these changes, residues R131[3.50] and Y218[5.58] of the DRY motif form an intra-helical ionic lock that further facilitates opening of the receptor binding pocket to engage the Gi protein. The Gαi subunit engages the receptor core using its amphipathic C-terminal α5 helix, forming polar interactions with CCR8 TM2, 3 and ICL3 and hydrophobic interactions with TM5 and ICL2 (Supplementary Fig. 10a, b). Comparison of the CCR8 · Giα5 interface to other CCR–Gi/o structures indicates high sequence conservation in this region and a similar mode of receptor engagement across the C-C chemokine receptor family (Supplementary Fig. 10c).

Mutation of residues identified from our structural analysis as likely important for CCL1-induced receptor activation or for both ligand binding and receptor activation confirmed their functional importance, with most mutants showing a complete or almost complete loss of receptor function (Fig. 4f, g). Notably, signal transmission in CCR8 relies on residues and activation motifs distinct from other CCRs. Unlike any other C-C chemokine receptor, CCR8 uses a Gln instead of the conserved Trp at position 2.60 to engage the CCL1 triad residue Ser25[46] (Supplementary Fig. 11). Highlighting the functional importance of this particular residue, the Q91W[2.60] mutation led to a dramatic decrease in CCL1-mediated activity[42], while we also observed strong functional impairment for mutant Q91A[2.60] (Fig. 4f). CCR8 also has a unique PMS activation motif instead of the canonical P[5.50]I[3.40]F[6.44] sequence, and is the only chemokine receptor that has a Ser instead of an aromatic residue at position 6.44 (Supplementary Fig. 11). Functional testing of CCR8 mutants M121I[3.40] and S247A[6.44] indicated that the presence of an Ile instead of a Met at position 3.40 is key for receptor activation, while position 6.44 appears more promiscuous (Fig. 4f).

## Binding mode and CCL1 blocking activity of mAb1

The binding of a chemokine to its cognate receptor is most often described by a two-step, two-site model, where the chemokine globular core first engages the receptor N-terminus at CRS1, followed by insertion of the chemokine N-terminus into the orthosteric core of the receptor at CRS2[47–49]. Recent studies have further refined this model, revealing a more complex interdependence of these regions in chemokine binding[40,50]. MAb1 inhibits CCL1 agonism of CCR8, but binds at an angle that positions the majority of the Fab away from the central cavity and limits the steric clash with CRS1, used by CCL1 for binding (Supplementary Fig. 12). This suggests the intriguing hypothesis that mAb1 may not block the initial binding of CCL1 at CRS1 but only prevents binding of CCL1 at CRS2, and would explain the disconnect we observe between the cell-based affinity and IC50 of CCL1 inhibition for mAb1.

To test this idea, we measured the real-time binding of fluorescently labeled human CCL1 (hCCL1[AF647]) to CHO.hCCR8 cells using LigandTracer[51,52]. We observed a biphasic binding profile consisting of a low affinity (-15 nM) binding event with fast on, fast off binding kinetics and a high affinity (-0.7 nM) binding event with slow on, slow off binding kinetics, with an overall affinity of ~1 nM (Fig. 5a, Supplementary Table 4), in agreement with previously reported affinities for unlabeled CCL1[53–55]. We next employed a time-resolved cell-based quenching assay[56,57], where we first bound hCCL1[AF647] to CHO.hCCR8 cells and then added unlabeled or quencher-labeled mAb1 (Fig. 5b). Addition of unlabeled mAb1 had no effect on the hCCL1[AF647] fluorescence signal, but addition of quencher-labeled mAb1 resulted in a decrease in signal over time, indicating that mAb1 is able to bind to a pre-existing hCCL1[AF647]-CCR8 complex on the cell. Taken together, our results support a two-site, two-step binding mechanism of CCL1 to CCR8, in which low affinity binding at CRS1 is followed by high affinity binding at CRS2 that requires conformational changes in the receptor. Furthermore, these studies demonstrate that mAb1 likely inhibits CCL1 activity by interfering with its binding to CRS2 on CCR8.

## Discussion

Antibodies targeting GPCRs hold great promise for therapeutic use, in particular for protein ligand GPCRs such as chemokine receptors that

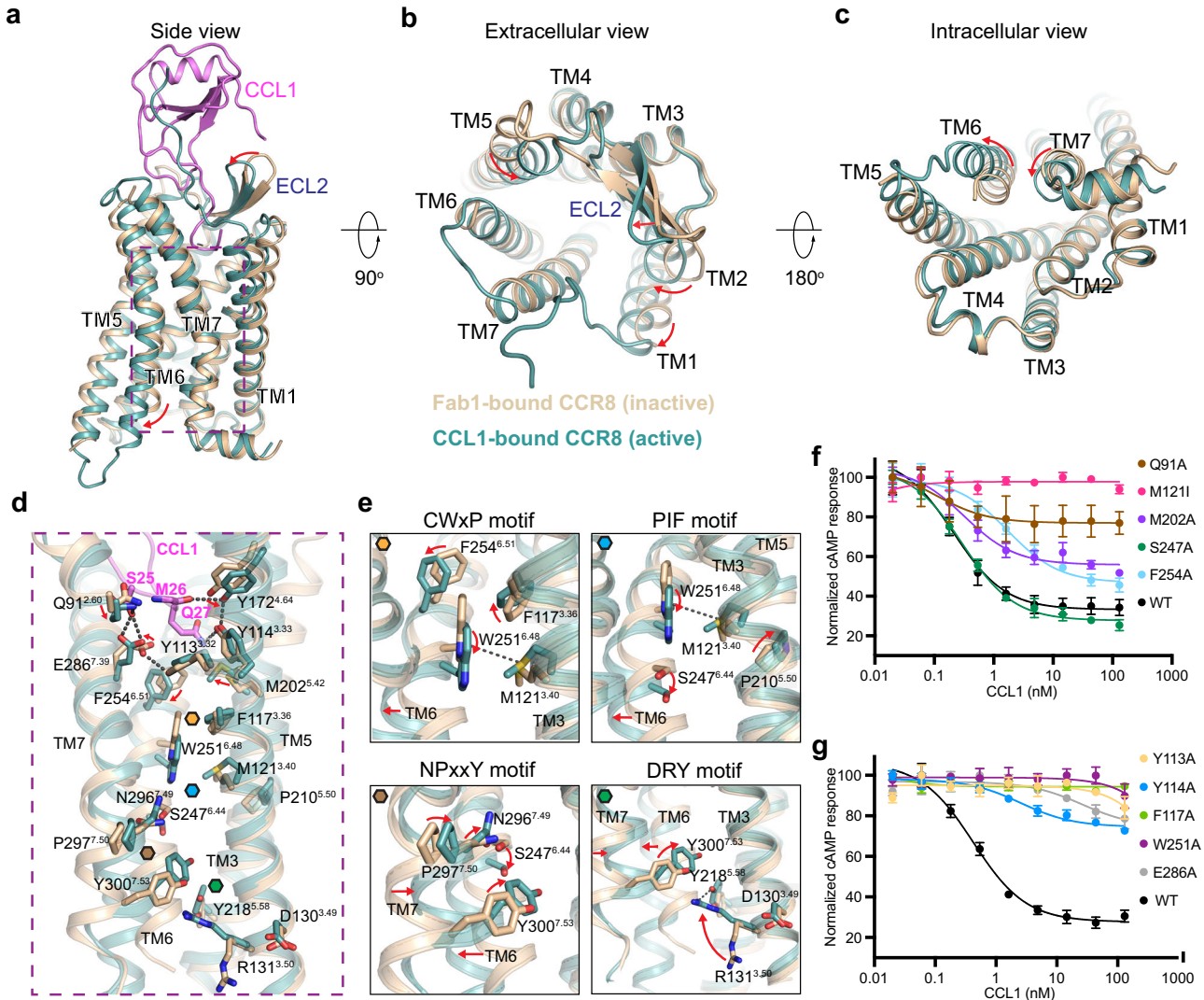

**Fig. 4 | CCL1-induced CCR8 activation and signal transduction.** Overlay of inactive and active CCR8 structures, colored in wheat and teal, respectively and shown as side view (**a**), extracellular view (**b**) or intracellular view (**c**), with major conformational changes indicated by red arrows. **d** Transmission of CCL1 binding signal to CCR8 activation microswitches, with the location of the CWxP, PIF, NPxxY and DRY motifs indicated by orange, blue, brown and green hexagons, respectively. **e** Close-up view of the CWxP, PIF, NPxxY and DRY motifs, with key side chain

rearrangements induced by CCL1 binding indicated by red arrows. **f**–**g** Normalized cAMP levels measured on HEK293T cells transiently expressing WT or mutated CCR8 in the presence of 1 μM Forskolin and increasing concentrations of CCL1. Data are representative of two independent experiments performed in triplicates (mean ± S.D). The best-fit IC50 and Imax values, along with relative receptor expression levels, are reported in Supplementary Table 2. Source data are provided as a Source Data file.

may be more challenging to drug with small molecules[40]. Recent studies are expanding our understanding of antibody features required for the efficient engagement of class A GPCRs and of antibody-receptor epitopes that allow antibodies to functionally modulate GPCRs. Here, our structural characterization of CCR8 bound to either an antagonist antibody or its endogenous agonist CCL1 revealed distinct chemokine-receptor engagement and activation features and rationalized how distinct antibody binding modes can inhibit CCR8 function (Fig. 5c).

Comparison of inactive-state and active-state CCR8 structures and analysis of the receptor-chemokine interactions allows us to rationalize how signal transmission from CRS2 to the canonical GPCR microswitches[44,45] could occur to enable the outward motion of the intracellular side of TM6, the hallmark of GPCR activation. CCL1 engages the CCR8 orthosteric core using a unique N-terminal conformation, where the most N-terminal CCL1 residue simultaneously engages two distinct receptor interfaces. Through polar interactions with CCR8 residues Y94[2.63], D97[2.66] and H283[7.36], CCL1 residue K24 acts

as an upper anchor on one side of the CCL1 S25-M26-Q27 triad, thereby pushing it towards the bottom of the receptor orthosteric pocket, while CCL1 residue P29, held in place through hydrophobic interactions with CCR8 residue Y184[ECL2], acts as a hinge on the other side of the triad (Fig. 3c, d). This positioning allows the triad residues to efficiently interact with CCR8 residues that are involved in receptor activation. In detail, S25 interacts with CCR8 residues E286[7.39] and Q91[2.60], M26 interacts with Y172[4.64] and Q27 interacts with Y114[3.33] and Y184[ECL2] (Fig. 4d). These interactions indirectly rearrange the TM3 aromatic connector residues Y114[3.33] and F117[3.36][26], and the TMs 6 and 7 residues F254[6.51] and F290[7.43], respectively (Fig. 4d). Together, these changes trigger a downward motion of the key toggle-switch activation residue W251[6.48] and its interaction with M121[3.40], thereby assisting TM6 relocation (Fig. 4e).

Our structural findings and mechanistic interpretation are in line with the dramatic functional impairments we observe for CCR8 mutants Y113A[3.32], F117A[3.36], M121I[3.40], and W251A[6.48] and the more modest or lack of functional impairment for mutants M202A[5.42],

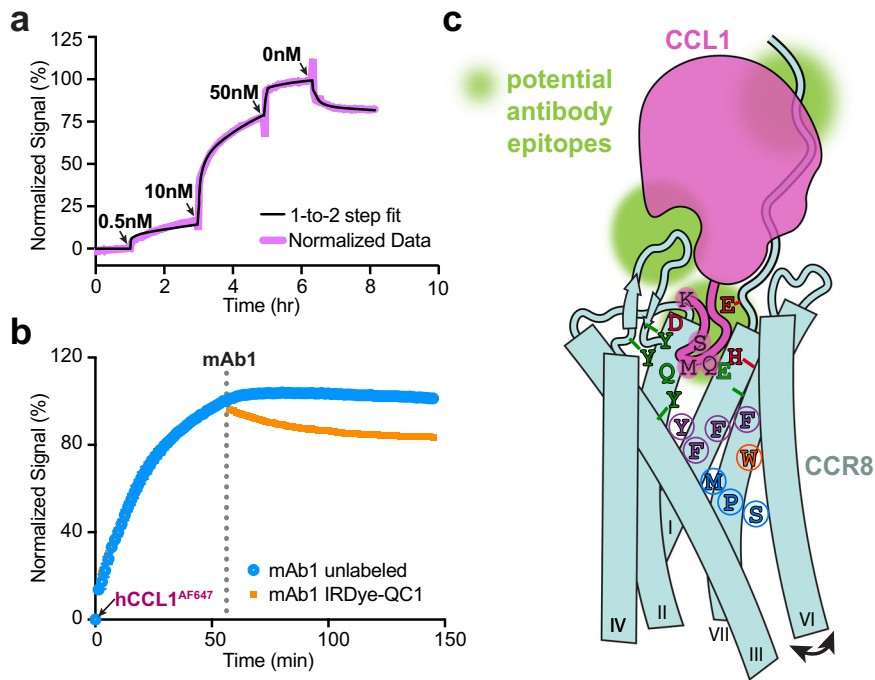

**Fig. 5 | CCL1 binding mode and mAb1 blocking activity. a** Representative real-time cell binding experiment measuring hCCL1$^{AF647}$ binding kinetics to huCCR8.CHO cells. The pink trace represents a normalized data trace and the black trace the calculated "one-to-two step" fit. The calculated average on- ($k_a$) and off-rates ($k_d$) for each step as well as affinities ($K_D$) from 3 independent experiments are listed in Supplementary Table 3. **b** Time-resolved cell-based quenching assay monitoring the dual binding of mAb1 and CCL1 to CCR8. The hCCL1$^{AF647}$ ligand is added at time 0 after which unlabeled mAb1 (blue trace) or IRDye-QC1 labeled mAb (orange trace) is added at 1 h. Data were normalized to each respective time point just prior to mAb1 addition. **c** Schematic summarizing key chemokine-receptor

interactions and activation switch residues that contribute to CCL1-induced CCR8 activation. The CCL1 is colored magenta, with key N-terminal residues $^{24}$KSMQ$^{27}$ shown as magenta circles. Important CCR8 residues interacting with the CCL1 anchor residue K$^{24}$ or the CCL1 triad residues $^{25}$SMQ$^{27}$ are shown in red and green, respectively. The proposed binding epitopes of antibodies are shown as green circles. CCL1 is colored magenta, with its N-terminal residues shown as black letters. The residues of CCR8 interacting with the N-terminus of CCL1 are shown as green letters. Key signal transmission residues are indicated in purple, orange (toggle switch) and blue (PIF motif). Source data are provided as a Source Data file.

F254A$^{6.51}$ and S247A$^{6.44}$ (Fig. 4f, g and Supplementary Table 2). Overall, our results agree with previous functional studies on CCR8, where alanine mutation of Y113$^{3.32}$, Y172$^{4.64}$, Y184$^{ECL2}$ and W251$^{6.48}$ also resulted in dramatic losses in potency[42]. These interpretations are also supported by previous mutagenesis studies of CCL1, where alanine mutation of K24, R32 and the triad residues S25-M26-Q27 led to the most dramatic reduction in signaling[58] (Supplementary Table 5).

Intriguingly, only five human CC chemokines contain basic residues in their N-terminal sequence preceding the CC motif (Supplementary Fig. 13). Of these, only CCL1 and CCL18 have been reported to be CCR8-specific, with CCL18 having a 10-fold lower affinity and a lower activity[53,59,60]. However, more recent studies failed to detect CCL18 binding to Jurkat cells stably expressing CCR8[53]. We tested whether CCL18 could induce a functional response in both of our cell-based assays, and while CCL1 induced a dose-dependent response as expected, we did not observe any response for CCL18 (Supplementary Fig. 14). Strikingly, the CCR8-specific viral chemokine antagonist MC148, with a reported affinity similar to CCL1[61,62], has a short N-terminus of only 5 amino acids, which contains a triplet of basic residues (Supplementary Fig. 13). We speculate that this feature could allow MC148 to efficiently engage CCR8, possibly through the same key acidic residues in CRS1.5 and CRS2.5 as CCL1, but without the ability to engage the receptor core, rationalizing why it would instead act as a potent antagonist.

While the existing therapeutic anti-GPCR antibodies bind to epitopes within the N-terminus, more examples of ECL-binding antibodies are emerging and may suggest common themes for binding. For

example, mAbs targeting HT2RB, CXCR4, and FPR1 possess long CDRH3s which form an essential β-strand/β-strand interaction in ECL2, as demonstrated by structural or modeling studies[11,63,64]. Intriguingly, our mAb1 recognizes CCR8 via a similar CDRH3-ECL2 interaction. This emerging theme could be further leveraged to build new in vitro antibody display libraries with long, structured CDRH3s to better engage the ECLs of class A GPCRs[65]. Additionally, the convex paratope of heavy chain only antibodies (VHHs), including their long CDRH3, appears ideally suited to engage the orthosteric pocket of GPCRs. This feature was recently illustrated by the structural characterization of an antagonist VHH against the apelin receptor that could be engineered into an agonist by inserting a Tyr in its CDRH3[14]. Together, these findings suggest that long CDRH3s may be one of the important features for developing effective antibody modulators against class A GPCRs.

Developing antibodies that engage different extracellular regions of a GPCR may allow to finetune the desired level of functional modulation. Early work using anti-CXCR2, -CXCR4, or -CXCR7 VHHs demonstrated that combining two VHHs against distinct extracellular epitopes into a single biparatopic molecule resulted in greatly improved inhibitory activities[66–68]. For the anti-CCR8 mAb1, the large difference we observe between its binding affinity and its IC50 of CCL1 inhibition suggests that engaging additional or other extracellular regions than ECL2 may yield stronger CCL1 blocking activity (Fig. 5c).

Taken together, our results provide additional insights into antibody-based targeting of class A GPCRs, and give a detailed molecular description of the mechanisms underlying CCR8 activation by

CCL1, providing a structure-function framework that might facilitate the development of future antibody therapeutics targeting chemokine receptors in particular and more broadly, class A GPCRs.

## Methods

### Generation of anti- human CCR8 mAb1

Anti-hCCR8 mAb1 was generated via immunization of New Zealand White rabbits (Charles River Laboratories, Hollister, CA) using established methods[69]. Animals used in these studies were maintained in an Association for Assessment and Accreditation of Laboratory Animal Care-accredited animal facility. All experiments were performed in compliance with Genentech's Institutional Animal Care and Use Committee (IACUC) and National Institutes of Health's Office of Laboratory Animal Welfare Guidelines. Approval of the study design was obtained from the Genentech IACUC prior to the start of this work. A humanized version of mAb1 was generated by inserting the CDR regions of the light and heavy chain domains into the closest human consensus germ line and grafting various Vernier positions onto the human germ lines. IgG and Fab expression constructs for the light chain and heavy chain for mAb1 were obtained by gene synthesis (Genscript, South San Francisco, CA). IgG and Fab were expressed by transient transfection of Expi293 cells (Expi293F, Invitrogen) and purified with affinity chromatography followed by SEC using standard methods (MabSelect SuRe or HiTrap Protein G; GE Healthcare, Piscataway, NJ, USA).

### Flow cytometry analysis of mAb1 binding to human Treg cells.
MAb1 was evaluated for binding to Treg cells by Fluorescence-Activated Cell Sorting (FACS) flow cytometry. Human colorectal DTC (Discovery Life Sciences) were thawed according to the vendor's protocol. Human PBMCs were isolated by Ficoll gradient centrifugation from buffy coats from healthy donors, collected as part of the Genentech blood donor program with written informed consent, and approval from the Western Institutional Review Board. Cells were stained with eFluor 780-conjugated Fixable Viability Dye (Thermo-Fisher Scientific) and 2 µg/mL mAb1, anti-OX40 (positive control, clone 3C8, produced in-house), anti-ERBB2 (negative control, clone 4D5, produced in-house), or anti-hIgG (negative control, produced in-house) for 20 min at 4 °C followed by secondary detection with AF647-conjugated AffiniPure F(ab')2 Fragment Goat anti-Human IgG, Fcγ fragment specific (109-606-006, Jackson ImmunoResearch, dilution: 1:500) for 10 min at 4 °C. To test the commercial anti-CCR8 antibody (clone 433H, BD Biosciences), cells were surface stained with eFluor 780-conjugated Fixable Viability Dye and APC-conjugated anti-Human CCR8 at a dilution of 1:100 in 100 µL. Cells were then intracellularly stained using the eBioscience Foxp3/Transcription Factor Staining Buffer Set (ThermoFisher Scientific) according to the manufacturer's protocol. Antibodies used to define T cell populations were CD45 (HI30, 1:200), CD3 BUV395 (SK7, 1:200), CD8 FITC (RPA-T8, 1:100) from BD Biosciences, CD4 BV421 (RPA-T4, 1:100) and CD14 PerCP-Cy5.5 (63D3, 1:400) from BioLegend and FOXP3 (236 A/E7, 1:25) from ThermoFisher Scientific. Flow cytometry was performed on a Fortessa X-20 (BD Biosciences) and analyzed with FlowJo software (BD Biosciences, Version 10.5.3).

### Flow cytometry analysis of antibody specificity.
Constructs encoding for CCR2-5, CCR8, CXCR4, ACKR2, and ACKR4 with an N-terminal FLAG tag were generated by gene synthesis (Genscript, South San Francisco, CA). The FLAG tag enabled the detection of cell surface expression of each GPCR. HEK293 cells were transfected with individual GPCR constructs or with a mock construct using transIT X2 (Mirus Bio LLC; Madison, WI, reagent:DNA = 3:1) for 24 h. Cells were then harvested and stained with mAb1 or rabbit anti-Flag pAb (F7425, Sigma-Aldrich; St. Louis, MO) at 5 µg/ml, followed by AF647-anti-hIgG or AF647-anti-RbIgG (109-606-170 or 111-606-144, Jackson

ImmunoResearch Laboratories Inc.; West Grove, PA, dilution: 1:500) respectively. Then, cells were washed twice with FACS buffer (phosphate-buffered saline containing 0.5% bovine serum albumin and 2 mM ethylenediaminetetraacetic acid) and resuspended in FACS buffer containing propidium iodide (BD Biosciences; 0.5 mg/mL) for analysis on a BD FACSCelesta™ Cell Analyzer (BD Biosciences; Franklin Lakes, NJ). Data were analyzed using FlowJo software (Version 10.6.1; FlowJo LLC; Ashland, OR).

### Radiolabeled mAb1 competitive binding assay.
Iodine-125 (125I) was stored as sodium iodine in 0.1 N sodium hydroxide (Perkin Elmer, Wellesley, MA). 1 mCi of [125]I was used to radiolabel random tyrosine residues on mAb1 using the indirect Iodogen method (Pierce). Radiolabeled [125]I-mAb1 was purified from unreacted iodine using a NAP5 column equilibrated with PBS. CHO.CCR8 cells were seeded in Opti-MEM supplemented with 2% FBS, 50 mM HEPES pH 7.2 and 0.1% Sodium Azide, at 50,000 cells per well. Unlabeled mAb1 starting at 50 nM was serially diluted 1:3 and mixed with a fixed concentration of [125]I-mAb1. The antibody mixture was added to the cells and incubated at room temperature on an orbital shaker for 18 h under gentle agitation. The cells and antibodies were then transferred to Millipore multiscreen filter plates. The filter plates were washed four times with 250 µL of cold binding buffer and dried for at least 30 min and the filters were punched into 5 mL polystyrene tubes. The radioactivity was measured using a Perkin Elmer Wallac Wizard 2470 Gamma Counter set at 1 count per minute with 0.8 counting efficiency. The data were fitted in GraphPad Prism (GraphPad Software, La Jolla, CA) using a heterologous one site-fit Ki competitive binding model. The competitive binding curve shown was performed in triplicate, with the same iodinated mAb1 tested in triplicate on the same batch of CHO cells stably expressing human CCR8. Data shown are means ± standard error of the mean of three technical replicates and are representative of three independent experiments. We did not observe significant variation between replicates that would have justified using additional replicates. The Ki measured across three independent experiments, each performed in triplicates, were 28.4 pM, 41.4 pM, and 55.5 pM. As mAb1 must be freshly iodinated before each experiment because of the decay of specific activity overtime, the concentration of radiolabeled Ab and the total cpm count may be slightly different between experiments, therefore we decided to not average the three experiments.

### Epitope mapping of mAb1 by flow cytometry.
Constructs encoding for human CCR8.CCR5 chimeras (N-term1, N-term2, ECL1, ECL2, and ECL3) in which different extracellular regions of CCR8 were replaced with the corresponding region from CCR5 with a C-terminal FLAG tag were generated. Each region is defined as follows based on the sequence in CCR8: N-term1 (Met1-Ser23), N-term2 (Met1-Lys35), ECL1 (Gln91-Val104), ECL2 (Tyr172-Lys193), and ECL3 (His264-Gly271). HEK293 cells were transfected with constructs encoding for chimeric hCCR8 or with a mock construct using transIT X2 (reagent:DNA = 3:1) for 24 h. Transiently transfected HEK293 cells were surface-stained with 5 µg/mL of anti-CCR8 mAb1 in FACS buffer at 4 °C for 30 min. After staining, the cells were washed twice with FACS buffer and stained with Alexa Fluor® 647 AffiniPure F(ab')₂ Fragment Goat Anti-Human IgG (109-606-006, Jackson ImmunoResearch Laboratories Inc.; West Grove, PA; dilution: 1:500) at 4 °C for 15 min. Then, the cells were washed twice, fixed and permeabilized with the BD Cytofix/Cytoperm Fixation/Permeabilization Kit (BD Biosciences), and stained with mouse monoclonal anti-FLAG M2-FITC antibody (F4049, Sigma Aldrich; St. Louis, MO; dilution: 1:100) at 4 °C for 30 min. Then, the cells were washed twice with permeabilization buffer and resuspended in FACS buffer for analysis on a BD FACSCelesta™ Cell Analyzer (BD Biosciences; Franklin Lakes, NJ). Data were analyzed using FlowJo software (Version 10.6.1; FlowJo LLC; Ashland, OR). Anti-CCR8 staining in the Flag positive population is shown.

**CCR8 calcium flux assay.** CCR8 activation was monitored by $Ca^{2+}$ influx using Fluorescent Imaging Plate Reader (FLIPR) FDSS/µCell (Hamamatsu, Japan). Briefly, CHO.hCCR8 cells were loaded with fluorescence $Ca^{2+}$ dye Fluo-8 NW (Cat#36307, AAT Bioquest) and incubated 30 min at 37 °C, and then at room temperature for another 30 min. For the CCL1, mAb1, isotype control or CCL18 binding assays, serially diluted hCCL1 (R&D systems), mAb1 (produced in-house), isotype control antibody (produced in-house) or CCL18 (R&D systems) were serially diluted in HHBS buffer in a clear 384-well plate. For the competition binding assay, serially diluted mAb1 was prepared in HHBS buffer in a clear 384-well plate and hCCL1 in HHBS buffer was aliquoted in a clear 384-well plate. The FLIPR assay was performed on FDSS/µCell with ligand addition at 10 s for the binding assay and mAb1/isotype addition at 10 s followed by hCCL1 addition at 300 s for the competition binding assay, with a total monitoring of 500 s. The excitation and emission wavelengths were set at 485 nm and 525 nm, respectively. After the run, a negative control correction was applied and the mAb1 data and CCL18 data were normalized to the hCCL1 signal (corresponding to 100%) and plotted as a function of mAb1 concentrations using GraphPad Prism (GraphPad Software, La Jolla, CA). mAb1-related experiments were performed in duplicate and CCL18-related experiments in quadruplicate, with data plotted as mean ± standard deviation.

**ERK phosphorylation assay.** HuCCR8.Jurkat cells were incubated in culture media with indicated concentrations of anti-CCR8 antibody for 30 min at 37 °C and then 20 nM CCL1 was added for 5 min. After stimulation, cells were harvested and washed twice with cold PBS. Cell lysate ($2 \times 10^5$ cells) were loaded on 8% pre-cast SDS-PAGE (Thermo Fisher Scientific) and analyzed by Western blot using antibodies specific for phospho-p44/42 MAPK (ERK1/2) (Thr202/Tyr204) (D13.14.4E), p44/42 MAPK (ERK1/2) (137F5) (Cell Signaling Technology), and GAPDH (AbD22549, Bio-Rad). Bands were visualized using an ECL detection system (Advansta). Bands were quantified using Image Studio Lite software and pERK values were normalized to the corresponding total ERK for each condition.

**Expression and purification of human CCR8 and CCL1–CCR8 fusion.** The CCR8 expression plasmid was constructed by inserting residues M1-L355 of human CCR8 into a mammalian expression vector. An HA signal sequence, Flag tag and TEV protease sequence were added to the N terminus of hCCR8, while a 3C protease sequence, GFP sequence and 2xStrep tag were inserted at the C terminus of hCCR8. The CCL1-CCR8 fusion construct was made by linking the C-terminus of full-length human CCL1, including its signal sequence, to the N-terminus of CCR8 residues D2-L355 using a 12x GS linker. To further stabilize the complex, we introduced a disulfide bridge by mutating residues Ala38 and Phe21 to cysteine in CCL1 and CCR8, respectively. A Flag tag, 3C protease sequence, GFP sequence and 2xStrep tag were inserted at the C terminus of hCCR8. Expi293F™ cells (ThermoFisher Scientific) were cultured in Expi293 expression medium at 37 °C, 8% $CO_2$. Cells were seeded at $2.5-3 \times 10^6$ viable cells per ml and transfected with 0.8 mg/ml DNA construct using the Expifectamine™ 293 transfection reagent (ThermoFisher Scientific). Post transfection, cells were fed with enhancers and 0.1 uM AZ-084 in DMSO (MedChemExpress) was added to the inactive-state cell culture only. The cells were harvested after 48 h, by centrifuging at $500 \times g$, 15 min, 4 °C. The cell pellet was resuspended in the lysis buffer A (25 mM HEPES, pH 7.5, 150 mM NaCl, 2 µM AZ084, 1´ complete protease inhibitor mixture; Roche). The suspension was added with 1% (w/v) Lauryl Maltose Neopentyl Glycol (LMNG; Anatrace), 0.1% (w/v) cholesteryl hemisuccinate TRIS salt (CHS; Anatrace) homogenized with a dounce homogenizer. After incubating at 4 °C for an hour, the suspension was ultracentrifuged with a 45Ti rotor ($100,000 \times g$, 1 h, 4 °C). The supernatant was applied to a Econo-Pac® Chromatography

Column (Bio-Rad Laboratories) packed with FLAG M2 affinity resins (Sigma-Aldrich) which was pre-equilibrated with buffer B (25 mM HEPES, pH 7.5, 150 mM NaCl, 0.01% LMNG, 0.001%CHS, 2 µM AZ084). The resins were washed by ten column volumes (CV) buffer B and eluted by 10CV buffer B containing 0.2 mg/ml FLAG peptide (Sigma-Aldrich). The eluent was further purified by a Superdex 200 10/300 column (GE Healthcare) with buffer B.

**Expression and purification of Fab1.** Fab fragments including Heavy and Light Chains of rabbit anti-human CCR8 mAb1 were transformed into 64B4 *E. coli* cells. After cell lysis, the lysate was clarified by centrifugation at $10,000\,g$ for one hour. The Fab was purified from the supernatant by passing through the affinity chromatography resin of KanCap™G (Kaneka) and cation exchange chromatography using SP HP Sepharose (Cytiva). The Fab was dialyzed into a buffer containing 25 mM HEPES, pH7.5, 150 mM NaCl.

**Expression and purification of scFv16.** ScFv16 was expressed and purified as described before[37]. Briefly, scFv16 was expressed by secretion from baculovirus-infected Trichoplusia ni (Hi5) cells. The filtered supernatant was loaded to an Ni-Sepharose (Qiagen) packed open-column which was equilibrated with 20 mM Hepes, pH 7.5, 300 mM NaCl and 5 mM imidazole. The column was washed with 20 mM Hepes pH, 7.5, 300 mM NaCl and 20 mM imidazole and eluted with 20 mM Hepes pH, 7.5, 300 mM NaCl and 300 mM imidazole. The eluted protein was further polished by size exclusion chromatography on a Superdex 200 16/60 column (GE healthcare) equilibrated with 20 mM Hepes, pH7.5 and 150 mM NaCl.

**Expression and purification of Gαi1 and Gβ1γ2 protein.** Gai1 was expressed in *E.coli*. The harvested cell pellet was solubilized in 50 mM Tris pH8.0, 150 mM NaCl, 10% glycerol, 1 mM Tris(2-carboxyethyl) phosphine hydrochloride (TCEP, Thermo Scientific), 5 mM imidazole and 1× complete protease inhibitor mixture (Roche). After cell lysis, the cell lysate was centrifuged at $5000 \times g$ for 50 min. The supernatant was passed through a Ni-Sepharose (Qiagen) packed open-column which was equilibrated with 50 mM Tris, pH8.0, 300 mM NaCl, 1 mM TCEP, 5 mM imidazole. The bound Gai1 protein was washed with 50 mM Tris, pH8.0, 300 mM NaCl, 1 mM TCEP, 20 mM imidazole and eluted with 50 mM Tris, pH8.0, 300 mM NaCl, 1 mM TCEP, 300 mM imidazole. The eluted protein was further subjected to a Superdex 200 16/60 column (GE healthcare) equilibrated with 50 mM Tris pH8.0, 150 mM NaCl and 1 mM TCEP.

The Gβ1γ2 heterodimer was expressed and purified as described before[70]. Briefly, Gβ1γ2 heterodimer was expressed in Hi5 insect cells. The cell pellet was solubilized in a hypotonic buffer containing 20 mM Hepes, pH7.5, 5 mM β-mercaptoethanol (β-ME) and 1x complete protease inhibitor mixture (Roche). After centrifugation, the collected cell membrane was solubilized with 20 mM Hepes, pH 7.5, 150 mM NaCl, 1% (w/v) sodium cholate, 0.05% dodecyl maltoside, 0.005% CHS (DDM/CHS Pre-Made Solutio, Anatrace), 5 mM β-ME and 5 mM imidazole overnight at 4 °C. The solubilized membrane solution was cleared by high-speed centrifugation and the supernatant was loaded onto a Ni-Sepharose (Qiagen) packed column. The column was washed with 20 mM HEPES, pH 7.5, 300 mM NaCl, 0.1% LMNG, 0.01% CHS, 5 mM imidazole, 1 mM TCEP and 20 mM HEPES, pH 7.5, 300 mM NaCl, 0.01% LMNG, 0.001% CHS, 20 mM imidazole, 1 mM TCEP. The bound Gβ1γ2 heterodimer was eluted with 20 mM HEPES, pH 7.5, 300 mM NaCl, 0.01% LMNG, 0.001% CHS, 300 mM imidazole and 1 mM TCEP. The eluted protein was mixed with TEV protease to cleave the N-terminal 6x His-tag and dialyzed overnight in 20 mM HEPES, pH 7.5, 150 mM NaCl, 0.01% LMNG, 0.001% CHS, 1 mM TCEP. The undigested Gβ1γ2 heterodimer was removed by reverse Ni-NTA affinity. The resulted cleaved Gβ1γ2 was then incubated for 1 h at

4 °C with lambda phosphatase (New England Biolabs), calf intestinal phosphatase (New England Biolabs) and Antarctic phosphatase (New England Biolabs) to dephosphorylate the protein. $G\beta_1\gamma_2$ was further polished by size exclusion chromatography on a Superdex 200 16/60 column (GE healthcare) equilibrated with 20 mM Hepes, pH 7.5, 150 mM NaCl, 0.01% LMNG, 0.001% CHS and 0.1 mM TCEP.

**Fab1-CCR8 and CCL1-CCR8-Gi complex formation.** The purified CCR8 protein was incubated with a twofold molar excess of anti-hCCR8 Fab1 on ice for 1 h. The formed complex was further polished over a Superdex 3.2/300 GL column (GE Healthcare) in buffer C (25 mM HEPES, pH 7.5, 150 mM NaCl, 0.002% LMNG, 0.0002% CHS, 2uM AZ084). To form CCL1-CCR8-Gi complex, Gi heterotrimer was first prepared by mixing purified $G\alpha_{i1}$ and $G\beta_1\gamma_2$ protein at a 2:1 molar ratio on ice for 1 h. Then, the purified CCL1-CCR8 fusion protein was added and mixed with a 1.5-fold molar excess of the formed Gi heterotrimer protein. Following the incubation at room temperature for 1 h, apyrase (NEB) was added and the reaction mixture was incubated overnight at 4 °C. The excess Gi protein was removed by loading the mixture to an open column packed with Flag resin (Sigma-Aldrich). The eluted CCL1-CCR8-Gi complex was mixed with scFv16 on ice for 1 h and further purified by size-exclusion chromatography on a Superdex 200 3.2/300 column in 20 mM Hepes, pH 7.5, 150 mM NaCl and 0.00075% (w/v) LMNG, 0.00025% glyco-diosgenin (GDN), 0.0002% (w/v) CHS (Anatrace).

**Cryo-EM sample preparation and imaging.** To prepare cryo-EM grids, Fab1-CCR8 complex (0.5–1 mg/ml) and CCL1-CCR8-Gi complex (1–2 mg/ml) were added to 300 Mesh 1.2/1.3R Au Quantifoil grids. After blotting for 2 s with 100% humidity, grids were plunged into liquid ethane by using a Vitrobot Mark IV (ThermoFisher). Images of Fab1-CCR8 complex and CCL1-CCR8-Gi complex were collected on a Titan Krios (ThermoFisher FEI) operated at 300 keV and equipped with a K3 direct electron detector with BioQuantum energy filter or a Falcon 4 with a Selectris, respectively. Full data collection parameters for each sample are shown in Supplementary Table 1.

**Cryo-EM data processing.** All motion correction and contrast transfer function (CTF) estimations were performed using MotionCor2 and Patch-Based CTF Estimation in cryoSPARC[71]. A total of 15,735 video stacks were collected for Fab1-CCR8 complex and a total of 572,092 particles were selected after several rounds of reference-free 2D classification from the total 5,928,998 picked particles in cryoSPARC. An ab initio 3D reference model was generated using cryoSPARC and further refined to 2.8 Å, which was subjected to 3D reference classification in Relion[72]. The particle projections from one out three classes were transferred back to cryoSPARC for further non-uniform and local refinement. For the CCR8-CCL1-Gi-scFv16 complex, 22,326 movies were collected. After 2D classification with 7,833,253 picked particles, the selected 509,493 particles were subjected to ab initio reconstruction and hetero-refinement in cryoSPARC. A 3.4 Å density map with 72,029 particles was obtained after non-uniform refinement in cryoSPARC. Subsequently, this 3.4 Å map together with a few junk maps were subjected to hetero-refinement with 7,833,253 in cryoSPARC. Multiple rounds of ab initio reconstruction and hetero-refinement yielded a final 2.96 Å density map with 201,761 particles, which was further used for focused refinement using a mask around CCR8-CCL1 or Gi-scFv16. Locally refined maps were combined into a CCR8-CCL1-Gi-scFv16 composite map using PHENIX "combine focused maps" to aid model building[73]. The gold-standard Fourier shell correlation (FSC = 0.143) and local resolution of CCR8-CCL1 and Gi-scFv16 maps were determined within cryoSPARC. The resolution (based on FSC = 0.5 criterion) of CCR8-CCL1-Gi-scFv16 composite map was calculated with phenix[73]. The detailed processing schemes are described in Supplementary Figs. 3, 4.

**Model building.** For the Fab1-CCR8 complex, the AlphaFold structure of human CCR8 (AlphaFoldDB: P51685), the light chain of the Fab crystal structure of transglutaminase 2-specific autoantibody 693-10-B06[74], and the heavy chain of the Fab NA884 crystal structure[75] were docked as a rigid body into the cryo-EM map by UCSF Chimera[76]. The model was refined over multiple rounds of model building in Coot[77] and real-space refinement in Phenix[73]. The model was finally validated using phenix.validation_cryoem[78] with built-in MolProbity scoring. For CCL1-CCR8-Gi-scFv16 complex, the published xray structure of CCL1 (PDB 4OIJ), the refined CCR8 structure from Fab1-CCR8 complex and Gi/scFV structure from the published CCR5-Gi complex (PDB 7O7F) were docked as a rigid body into the cryo-EM map by UCSF Chimera. The CCL1-CCR8-Gi-scFv16 model was built in the same way as the Fab1-CCR8 complex by performing iterative rounds of real-space refinement in Phenix and model building in Coot. UCSF Chimera[76], UCSF ChimeraX[79], and PyMOL (PyMOL Molecular Graphics System, Version 2.4.1 Schrödinger, LLC.) were used to generate figures. The structural refinement statistics are provided in Supplementary Table 1.

**Molecular dynamics simulations.** The CCL1-CCR8-G-protein complex was simulated by extracting the model of the complex and mutating the engineered disulfide back to the native amino acids in PyMOL (CCR8 C21F; CCL1 C38A). The bound scFv16 was excluded from the system. The complex was oriented for insertion into a lipid bilayer using the OPM database entry for the CCR5-G-protein structure[27,80]. The simulation system was independently prepared in OpenMM[81] using pdbfixer by adding a POPC membrane totaling 476 molecules and solvated with 100 mM NaCl in a 12.7 nm × 13.4 nm × 18 nm simulation box, resulting in a total system size of 298,653 atoms (Supplementary Table 3). Four replicates of this system were then independently energy minimized to a tolerance of 10 kJ/mol, equilibrated at constant volume for 100 ps while increasing the temperature to 300 K, and then simulated isobarically at 1 bar. GaMD simulations were run using the gamd-openmm package available at https://github.com/MiaoLab20/gamd-openmm[43,82].

**Cell-based enzyme-linked immunosorbent assay (ELISA).** HEK293T cells transiently transfected to express wild type or mutated C-terminal Flag-tagged CCR8 were seeded at 100,000 cells per well of a 96-well plate. Cell-based ELISA was performed to quantify Flag-tagged receptors as previously described (Maurel et al.[83]). Briefly, cells were fixed with 4% paraformaldehyde, washed twice with phosphate-buffered saline (PBS), permeabilized with 0.5% triton for 5 min, washed twice with PBS and blocked with phosphate-buffered saline +1% FBS. After 30 min, receptor expression was detected using an anti-Flag-M2 monoclonal antibody conjugated with horseradish peroxidase (Sigma-Aldrich). After washes, bound antibodies were detected by chemiluminescence using SuperSignal substrate (Pierce) and a CLARIOstar reader (BMG LabTech).

**GloSensor cAMP inhibition assay.** HEK 293T were transfected with DNA encoding wild type or mutated C-terminal Flag-tagged CCR8 and the luciferase-based cAMP Glosensor-22F (Promega), using Lipofectamine 2000 (ThermoFisher Scientific) per manufacturer's instructions. 24 h later, the transfection media was discarded and the cells were incubated for 2 h in 90 µL of equilibration medium containing CO2 independent medium (Gibco), 10% FBS and 2% GloSensor cAMP reagent stock solution (Promega). 10 µL of 10 × 1:3 serially diluted CCL1 (Novus Biologicals) were added to the cells for 10 min prior to addition of 10 µL of 11X Forskolin (Fsk) (1 µM final) (MilliporeSigma). Plates were then incubated for 60 min at room temperature and luminescence was measured using the GloMax reader (Promega). Data was normalized as follows: 100% maximum signal, 0% media control and fit using the log(inhibitor) vs. response (three parameters) equation in Graphpad Prism v9. To compare relative expression levels and functional

properties across constructs, untransfected and wild-type controls were included in each dataset.

**Real-time cell binding experiments.** CHO wild type or CHO.hCCR8 cells were seeded at $1 \times 10^6$ cells in a $2 \times 2$ MultiDish (Ridgeview Instruments AB, Uppsala, Sweden) in section A and C, with wild-type CHO cells in B and D. Cells were incubated in culture medium (RPMI supplemented with 10% FBS, GlutaMAX, and Pen/Strep) overnight at 37 °C and 5% $CO_2$ overnight. Binding of AlexaFluor647-labeled hCCL1 (hCCL1$^{AF647}$; Almac, Scotland, UK, Catalog #: CAF-07) to cells was monitored using a LigandTracer® Green (Ridgeview Instruments AB) fitted with a red (632 nm)−NIR (671 nm) detector at room temperature the following day, similar to what has previously been described[52,84]. Briefly, the cell culture media was aspirated and replaced with room-temperature $CO_2$-Independent media (Gibco) supplemented with 10% FBS, GlutaMAX, and 0.02% sodium azide (Assay Media). The plate was then fitted into the LigandTracer® stage, and allowed to reach a stable baseline for 20-30 minutes. hCCL1$^{AF647}$ was added to the media in three sequential association steps at 0.5, 10, and 50 nM concentrations. Association of each step was measured until clear curvature could be observed, ~1.5−2 h each. After the final association step, the media was aspirated and media was added to measure the dissociation of hCCL1$^{AF647}$. The signal from CHO wild-type cells was subtracted from CHO.hCCR8 cells to correct for non-specific binding. Kinetic traces were analyzed using TraceDrawer software (Ridgeview Instruments), and fitted using a one-to-two step model.

**Time-resolved cell-based quenching assay.** Quenching experiments were performed as described previously[56]. Anti-CCR8 mAb1 was labeled with IRDye-QC1 (LiCOR Biosciences, Lincoln, NE, Catalog # 929-70030) according to the manufacturer's instructions. Cells and assay media were prepared as described above. Once cells reached a stable baseline (~30 min), hCCL1$^{AF647}$ was added to the media at a concentration of 10 nM and association was observed for ~1 h, at which point unlabeled or IRDye-QC1 labeled mAb1 was added to the media at equimolar concentrations (10 nM). The fluorescence was measured for an additional 2 h. Data were normalized to the final time point prior to mAb1 addition and analyzed using GraphPad Prism version 9.4.1. Three independent experiments taken from distinct samples were performed with no technical replicates, but each experiment consisted of running the quencher labeled and unlabeled Ab1 in parallel. At the final time point, the mean normalized signal ± standard deviation was $101.3 \pm 1.19$ and $83.53 \pm 2.00$ for Unlabeled Ab1 and IRDye-QC1-labeled Ab1, respectively. No sample-size calculations were performed. The sample size was chosen due to very low variability across samples and time-sensitive experimental set up.

### Reporting summary
Further information on research design is available in the Nature Portfolio Reporting Summary linked to this article.

## Data availability
All data included in the paper and the supplementary information files are available. Sequence information of CCR8 and CCL1 was obtained from the Uniprot database (CCR8: P51685, CCL1: P22362). Source data are provided with this paper for Fig. 1a–c, Fig. 3e, f, Fig. 4f, g, Fig. 5a, b, Supplementary Fig. 2c, e and Supplementary Fig. 14a,b. The 3D cryo-EM density maps have been deposited into the Electron Microscopy Data Bank under accession codes EMD-41370 (Fab1-CCR8), EM-41827 (Gi-scFv16), EM-41828 (CCL1-CCR8), EM-41829 (composite map of CCL1-CCR8-Gi-scFv16) and EM-41850 (consensus map of CCL1-CCR8-Gi-scFv16). The structure coordinates have been deposited in the PDB under accession codes 8TLM (Fab1-CCR8 structure) and 8U1U (CCL1-CCR8-Gi-scFv16 structure). Source data are provided with this paper. Initial coordinate and simulation input files as well as a coordinate files

of the final output of the GaMD simulations have been deposited in an open public repository [https://zenodo.org/records/10038937]. Source data are provided with this paper.

## Code availability
All MD simulations were performed and analyzed using the publicly available software tools OpenMM v7.7 [https://github.com/openmm/openmm], GaMD-OpenMM [https://github.com/MiaoLab20/gamd-openmm] and MDTraj [https://github.com/mdtraj/mdtraj].

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

## Acknowledgements

We would like to thank the Department of Biomolecular Resources for cloning and expression support, the Antibody Production and Automation Technologies (APAT) group for antibody purification support and Joyce Chan for help with cAMP assays.

## Author contributions

Y.Sun and C.C. established immunization protocols, generated and characterized mAb1. Y.Sun performed flow cytometry and FACS experiments. E.J. performed and analyzed LigandTracer and Time-resolved cell-based quenching assays. T.Z. performed and analyzed cAMP and ELISA assays under supervision of L.C-A. L.C-A. supervised and analyzed the radiolabeled mAb1 competitive binding assay. J.M.N. performed, analyzed and interpreted M.D. simulations. A.S. and T.L.A. generated stable cell lines under supervision of S.R. Y.Shi and J.W. developed, performed and analyzed CCR8 activity assays. B.W. performed pERK signaling assays. C.T. provided key molecular cloning support and P.J. provided key protein expression support. J.P and M.M. designed constructs for immunization and expression. D.S. and H.H. established protein purification protocols. M.J., C.A., and S.N. optimized cryo-EM sample preparation and performed cryo-EM data collection. D.S. processed cryo-EM data and built the model. D.S., J.T.K., and M.M. wrote the manuscript with input from all authors. S.R., J.T.K., and M.M. supervised the project and are co-senior authors.

## Competing interests

All authors are current or previous employees of Genentech Inc./Roche.
