## [Peer Review File · Nature Communications]

Structural basis of antibody inhibition and chemokine activation of the human CC chemokine receptor 8REVIEWER COMMENTS

Reviewer #1 (Remarks to the Author):

This study describes a structure-activity framework of the mechanism underlying CCR8, which providing an insight into antibody therapeutics targeting chemokine receptors. It appears that both structures are well resolved as shown in the figures. However, the accession ID of either density map or coordinates is not presented in the section of Data Availability. At least the validation report output from the wwPDB deposition should be provided. Here I list a few issues that should be addressed by the authors.

Major points:

1. To validate the key binding sites between CCL1 and CCR8, site-directed mutagenesis of the key recognition sites on CCR8 should be performed. Then cAMP inhibition or calcium mobilization of CCR8 stimulated by the ligands should be carried out, as well as the expression levels of CCR8 mutants. After the functional assays, whether the recognition mechanism of CCR8 is consistent with the previously reported mutagenesis data can be concluded.
2. The authors did a 200-ns molecular dynamics simulation to learn the conformational dynamics of CCL1 and CCR8 ECL3 in the CCL1-CCR8 complex. But 200 ns may be too short to confirm the interaction between the chemokine and the receptor. It will be more robust for a time scale longer than 1 μ s in the MD simulation.
3. As the authors have performed MD simulation of CCL1 interacting with CCR8, MD simulation of Fab1 bound to CCR8 should be performed. So that the conformation dynamics underlying the mechanism of CCR8 activation can be elucidated more deeply.

Minor points:

1. The mass of $G\alpha$ is larger than $G\beta$. But the position of $G\beta$ is higher than $G\alpha$ as shown in supplementary Fig 4. The authors should check again the labels in the SDS-PAGE gel image. In addition, the band of scFv16 has been shown in the SDS-PAGE gel image, so the peak of scFv16 also should be displayed in the SEC elution profile.
2. In supplementary Fig 5b, the cryo-EM density map of $\alpha 5$ helix of G_i protein should be shown.
3. In supplementary Fig 10c, the CCR8- G_i interface is compared with other receptors of CCR subfamily. The residues presented in sticks should be labeled.
4. In supplementary Fig 7b, the structure of human 5-hydroxytryptamine 2B (5-HT2B) was used for comparison. Could the authors explain the reason for choosing 5-HT2B in the comparison?

5. The references (36, 40, 41) of previously defined chemokine recognition sites mentioned in line 161 should be revised. The authors can either refer to the CCR8 homologue such as CCR1, CCR5, and CCR6, or cite review articles summarizing the recognition modes of chemokine receptors.
6. In line 176-177, the authors stated that a previously unobserved feature in CCR8 is the CCL1 N-terminus folds back up away from the receptor core towards the extracellular regions. I am not quite sure about this conformation since the coordinates of CCL1-CCR8-Gi is not shown in the revision materials. But as shown in supplementary Fig 8, the pose of CCL20 in its N-terminus bound to CCR6 is similar to that of CCL1 in CCR8, showing a “hook-like” conformation. The author may need to compare the binding pose of CCL1 and CCL20 in more details.
7. In line 213-214, the description of interactions directly and indirectly rearranges Y1133.32, Y1143.33, and F1173.36 side chains is not clear. It will be more readable to clarify which residues form direct interaction and which residues mediate indirect interaction.
8. In the study of epitope mapping of mAb1 by flow cytometry, why did the authors select CCR5 for the design of constructs encoding for human CCR8-CCR5 chimeras?
9. In the legend of Fig 1d, “CCL1-CCR8-Gi1” should be changed to “CCL1-CCR8-Gi1-scFv16”.
10. In the legend of Fig 5c, what is the meaning of xx circles (line 846)?

Reviewer #2 (Remarks to the Author):

In their manuscript Sun et al describe the detailed structure of chemokine receptor CCR8 conjugated to its ligand CCL1. In addition, they describe the production of a specific antibody to CCR8 that inhibits CCL1-induced calcium signaling through CCR8. The antibody does not bind to the N-terminus of the receptor, like many other described anti chemokine receptor antibodies, but has a long CDRH3 that allows it to interact with the extracellular loops (mainly ECL2). This might be compared to nanobodies that have the advantage to bind better to epitopes that are oriented more in depth in their target proteins. Interestingly, the authors describe a unique orientation of the N-terminus of CCL1 when bound to CCR8 compared to other chemokine ligand receptor pairs. This unique orientation of the N-terminal residue creates a chemokine recognition site on the receptor (CRS2.5) which is so far unique in the chemokine – chemokine receptor interactions.

Main comments:

1. A unique biological characteristic of the CCL1 – CCR8 pair is that CCL1, in addition to its typical chemokine activities such as chemotaxis and induction of calcium signals, also has anti-apoptotic activity on T cells. This is in particular important in the context of the role of Treg cells in the tumor environment. Authors show that calcium signaling through CCR8 is inhibited by their antibody even though the antibody is still able to bind to a receptor-ligand pair. Is also the anti-apoptotic activity of CCL1 inhibited by the antibody. This would be highly interesting with regards to an application potential of the antibody.

2. There is discussion in the field on the interaction between CCL18 and CCR8. Can the authors conclude anything from their structural analyses on the potential of CCL18 to bind (or not to bind or activate) CCR8?

Minor comments:

1. Suppl. Fig. 7a: check the ligand receptor pair for CCR5 in the figure and for CCR2 in the legend. They are probably wrong.
2. Line 280: are these losses in potency correct? I don't find them in the Suppl. Table.
3. Correction of some typo's: Line 204: CCL1's instead of CCL's; line 541: delete "between"; Suppl Table 4: correct PMBC to PBMC

Reviewer #3 (Remarks to the Author):

The manuscript by Sun et al describes structure/function studies of CCR8. Two experimental structures are determined: 1) CCR8 in complex with Gi and agonist ligand CCL1 (which is engineered to bind CCR8 covalently via disulfide bridge); and 2) CCR8 in complex with antagonist antibody Fab1. By comparing these two structures and performing real-time binding assay the authors conclude that CCL1 binding to CCR8 follows a two-step mechanism, whereby low affinity binding at CRS1 is followed by high affinity binding at CRS2 that requires conformational changes in the receptor.

The manuscript also reports on computational MD simulations of CCL1-CCR8-Gi complex with and without the engineered disulfide bond between the ligand and the receptor. For each construct two replicates were run for relatively short, 250ns time. The goal of these simulations was to confirm that the conformational dynamics of CCL1 was not affected by the disulfide. The observation from the simulations was that removal of the engineered disulfide allowed the CCR8 N-terminus to move away from CCL1, but did not alter the conformational heterogeneity of CCL1, suggesting that the engineered disulfide does not affect how CCL1 engages the receptor core. First, this study and conclusion seems to be somewhat disconnected from the rest of the manuscript. How does this inference add to the experiments that are reported? Perhaps, some mechanistic predictions could have been made from these simulations and tested experimentally? Secondly, it seems to me that to make such strong conclusions, one needs to simulate the system for much longer time than 250ns. In general, the authors should make a better effort to present the significance of the presented simulations.

Overall, lot of structural comparisons are made, based on the cryo-EM models. The authors should be careful that these are just isolated structures and therefore inferring on some dynamic mechanisms from such comparisons could be misleading. For example, there is a sentence: "Careful analysis of the receptor-chemokine interactions allows us to rationalize how signal transmission from CRS2 to the canonical GPCR microswitches occurs to enable the outward motion of the intracellular side of TM6, the hallmark of GPCR activation." Seems like that the authors are describing very complicated allosteric

mechanism from simple structural comparison. I would advise against this. It is best just to use simple comparative description stating what is different between the two structures.

Response to Reviewers

We thank all our Reviewers for their support of this work and for their thoughtful evaluation and constructive feedback. The Reviewers' feedback has been very valuable to help us further elevate the presentation and impact of our work. New data and improvements to strengthen our revised manuscript are summarized here - in brief, we have:

1. Provided full validation reports for our structures as well as additional cryo-EM density maps for key regions in our structures.
2. Performed extensive functional mutagenesis studies to expand our structure-function analysis of ligand binding and receptor activation and to complement previously reported data.
3. Assessed the anti-apoptotic activity of our antibody on T cells and its effect on the MAPK-signaling pathway, demonstrating that CCL1 does not have anti-apoptotic effects in human T cells but that mAb1 reduces CCL1-induced ERK phosphorylation in a dose-dependent manner.
4. Completed additional Gaussian-accelerated Molecular Dynamics simulations of the CCL1-CCR8-Gi complex to strengthen our conclusion that the disulfide-engineering approach we used to stabilize the ligand-receptor assembly did not alter the overall structure or the conformational flexibility of the complex.
5. Investigated the potential interaction between CCL18 and CCR8 through two different cell-based assays and through computational structural biology methods, demonstrating that our data and analysis does not provide convincing evidence for CCL18 being a CCR8 ligand.
6. Revised our structural analysis of the receptor activation mechanism to avoid overinterpretation of the underlying dynamics or allosteric mechanisms by clearly separating experimental observations in the Results section from mechanistic interpretations in the Discussion section.

We hope that our Reviewers agree that we have significantly improved the work and that the revised manuscript merits consideration of timely publication. Accordingly, we have provided a point-by-point response to Reviewers below.

Reviewer #1 (Remarks to the Author):

This study describes a structure-activity framework of the mechanism underlying CCR8, which providing an insight into antibody therapeutics targeting chemokine receptors. It appears that both structures are well resolved as shown in the figures. However, the accession ID of either density map or coordinates is not presented in the section of Data Availability. At least the validation report output from the wwPDB deposition should be provided. Here I list a few issues that should be addressed by the authors.

We thank the Reviewer for their evaluation of our work. We have now deposited our structures to the PDB and provide the validation reports for both structures as part of the submission of our revised manuscript.

Major points:

1. To validate the key binding sites between CCL1 and CCR8, site-directed mutagenesis of the key recognition sites on CCR8 should be performed. Then cAMP inhibition or calcium mobilization of CCR8 stimulated by the ligands should be carried out, as well as the expression levels of CCR8 mutants. After the functional assays, whether the recognition mechanism of CCR8 is consistent with the previously reported mutagenesis data can be concluded.

We appreciate the Reviewer's suggestion, and at their request we have now generated our own functional mutagenesis data, which we now include in the manuscript to strengthen our interpretations beyond what was already reported in the literature. We selected a total of 19 key CCR8 residues to mutate, which based on our structural analysis and functional interpretation are either likely directly involved in CCL1 binding (D26, E28, E177, D178, Q182, Y184, H267, H283), likely to be important for CCL1-induced receptor activation (Y113, F117, M121, M202, S247, W251, F254), or involved in both ligand binding and

receptor activation (Q91, Y114, Y172, E286). We mutated each of these residues to alanine to disrupt these interactions, except for the P^{5.50}I^{3.40}F^{6.44} motif residue M121^{3.40}, which we mutated to the corresponding isoleucine residue found in CCR1, 2, 3, 4 and 5. Overall, we observe that our functional mutagenesis data supports our structural analysis of CCL1 binding and CCL1-induced activation. As expected, we observe that single alanine mutation of CCR8 residues only involved in ligand binding display increased IC₅₀ values and decreased maximal responses for CCL1-induced cAMP inhibition, but that overall, these effects are less dramatic compared to mutating residues that are part of signal transmission and/or key activation microswitches.

Of those residues only involved in ligand binding, mutations D26A^{Nterm}, E28A^{1.25}, E177A^{ECL2}, H267A^{6.64} show modest functional impairment compared to WT, while D178A^{ECL2}, Q182A^{ECL2}, Y184A^{ECL2} and H283A^{7.36} show stronger functional impairment compared to WT. These results clearly indicate that the CCR8 ECL2 is an important region for the stable engagement of CCL1 through interactions with the globular core and upper N-terminus of the ligand. In addition, the CCR8-CCL1 Y184^{ECL2} - P29 hydrophobic interactions and the H283^{7.36} - K24 sidechain-backbone interactions help lock the CCL1 S25-M26-Q27 triad in place. These receptor interactions with the lower N-terminus of CCL1 allow for its efficient interaction with CCR8 residues that are involved in both ligand binding and receptor activation (Q91^{2.60}, Y114^{3.33}, Y172^{4.64} and E286^{7.39}), for which individual alanine mutants all show dramatic functional impairment, validating our interpretation of their key respective roles.

For the residues which we identified as likely important for CCL1-induced receptor activation but that are not involved in ligand binding, we observe dramatic functional impairments for mutations Y113A^{3.32}, F117A^{3.36}, M121^{3.40} and W251A^{6.48}, while other residues we tested showed more modest (M202A^{5.42}, F254A^{6.51}) or no (S247A^{6.44}) functional impairment. This confirms that sidechain rearrangements and signal transmission from residues directly involved in ligand binding to the key highly conserved toggle-switch residue W251A^{6.48} are relayed mainly through the TM3 aromatic connector Y113^{3.32} - Y114^{3.33} - F117A^{3.36} (as described for CCR5 in Isaikina et al Sci Adv 2021), and less so through TM5/6 residues M202A^{5.42} and F254A^{6.51}. Our mutagenesis data also indicates that for the P^{5.50}I^{3.40}F^{6.44} motif residues M121^{3.40} and S247^{6.44}, the presence of an Ile instead of the more common Met at position 3.40 is key for receptor activation in CCR8, while position 6.44 appears promiscuous in CCR8, in line with the observed sequence conservation in the C-C chemokine receptor family for microswitch residues shown in **Fig. R1**.

Figure R1 Sequence conservation at C-C chemokine receptor microswitch residues. Alignment generated using the www.gpcrdb.com sequence analysis tool.

We now include these additional data in the revised manuscript in **Figures 3e & 4f** and **Supplementary Table 2** and have extensively updated the corresponding results and discussion sections of the manuscript, see lines 190-195, 232-244 and 280-303.

Figure R2 (now included as Figure 3e & 4f) Functional assessment of CCR8 mutants. cAMP levels were measured on HEK293T cells transiently expressing WT or CCR8 mutants in the presence of 1 μ M Forskolin and increasing concentrations of CCL1. Data were normalized as follows: 100% maximum signal, 0% media control. Data are representative of two independent experiments performed in triplicates (mean \pm S.D). To compare relative expression levels and functional properties across constructs, untransfected and wild-type controls were included in each dataset. Data was fit using the log(inhibitor) vs. response (three parameters) equation in Graphpad Prism v9. The best-fit IC50 and I_{max} values, along with relative receptor expression levels, are reported in **Supplementary Table 3**.

Supplementary Table 2: Effect of CCR8 mutations on receptor expression levels and CCL1-induced CCR8 signaling activity

	D26A	E28A	E177A	D178A	H283A	WT
% WT expression (mean ± SD)	98 ± 5	86 ± 26	123 ± 16	93 ± 16	106 ± 2	100 ± 22
Imax	32.4	36.95	34.58	53.87	45.43	27.42
Imax 95% CI	28.86 to 35.83	33.13 to 40.62	31.42 to 37.65	50.94 to 56.73	41.99 to 48.71	24.35 to 30.40
Imax fold-change over WT	1.2	1.3	1.3	2.0	1.7	
IC50	0.9822	1.071	0.9341	0.4145	2.832	0.5978
IC50 95% CI	0.7081 to 1.366	0.7227 to 1.589	0.6820 to 1.281	0.2657 to 0.6493	1.976 to 4.035	0.4406 to 0.8133
IC50 fold-change over WT	1.6	1.8	1.6	0.7	4.7	
	Q91A	M121I	M202A	S247A	F254A	WT
% WT expression (mean ± SD)	110 ± 26	97 ± 15	113 ± 11	121 ± 15	104 ± 22	100 ± 26
Imax	76.97	N/A	55.88	27.7	46.82	33.27
Imax 95% CI	72.57 to 81.11		52.90 to 58.76	25.71 to 29.67	42.75 to 50.65	30.08 to 36.42
Imax fold-change over WT	2.3		1.7	0.8	1.4	
IC50	0.1081	N/A	0.3202	0.2987	1.726	0.2113
IC50 95% CI	0.009774 to 0.6022		0.1967 to 0.5227	0.2401 to 0.3714	1.083 to 2.755	0.1480 to 0.3010
IC50 fold-change over WT	0.5		1.5	1.4	8.2	
		F117A	Y172A	Q182A	Y184A	WT
% WT expression (mean ± SD)		82 ± 21	77 ± 20	104 ± 10	99 ± 11	100 ± 8
Imax		N/A	82.31	42	42.72	31.91
Imax 95% CI			79.86 to 84.56	35.31 to 48.12	35.87 to 48.48	29.54 to 34.23
Imax fold-change over WT			2.6	1.3	1.3	
IC50		N/A	1.333	2.077	13.62	0.2509
IC50 95% CI			0.4226 to 3.624	1.100 to 3.946	8.735 to 21.37	0.1879 to 0.3346
IC50 fold-change over WT			5.3	8.3	54.3	
	Y113A	Y114A	W251A	H267A	E286A	WT
% WT expression (mean ± SD)	65 ± 9	79 ± 16	81 ± 18	104 ± 19	130 ± 17	100 ± 29
Imax	N/A	74.54	N/A	46.53	74.43	27.67
Imax 95% CI		70.32 to 78.11		43.42 to 49.58	48.86 to 82.53	24.26 to 31.02
Imax fold-change over WT		2.7		1.7	2.7	
IC50	N/A	2.547	N/A	0.2684	26.05	0.4218
IC50 95% CI		0.9933 to 6.736		0.1779 to 0.4035	6.430 to 173.9	0.3046 to 0.5834
IC50 fold-change over WT		6.0		0.6	61.8	

Table R1 (now included as Supplementary Table 2). Effect of CCR8 mutations on receptor expression levels and CCL1-induced CCR8 signaling activity.

Cell-based anti-Flag ELISA experiments were performed to determine the expression level of each C-terminally Flag-tagged construct transiently expressed in HEK293T cells and normalized relative to untransfected cells and a wild-type CCR8 control included in each dataset. Data are representative of two independent experiments performed in triplicates. Best-fit values and 95% confidence intervals for Imax and IC50 were calculated from the CCL1 - cAMP dose-response curves reported in Figures 3e and 4f using the log(inhibitor) vs. response (three parameters) equation in Graphpad Prism v9. Data are representative of two independent experiments performed in triplicates. Mutations showing major functional effects on Imax (> 1.5-fold difference over WT) and/or IC50 (> 2-fold difference over WT) are highlighted in red.

Of the 19 mutants we tested, 10 were never tested before (D26A, E28A, F117A, M121I, E177A, D178A, Q182A, M202A, S247A, H267A) while 9 have previously been reported in Jensen et al BJP 2012 (Q91A, Y113A, Y114A, Y172A, Y184A, W251A, F254A, H283A, E286A). We note that in Jensen et al, functional testing was done by measuring IP3 accumulation in transiently transfected COS-7 cells, whereas we measured cAMP inhibition in transiently transfected HEK293 cells. We also report the full dose-response curves and calculated both IC50 and I_{max} values, whereas the Jensen data only reported EC50 values, but no E_{max} values. We also note that many mutants showed <30% of WT expression in the Jensen study, which makes it more difficult to meaningfully compare these to WT, while in our study, all mutants showed more robust expression (>=65% of WT). The overall trends however in terms of the degree of functional impairment each respective mutant causes agree well between both datasets, as shown below in **Table R2**. Due to the differences described here, and since we have now generated our own extensive set of mutations to accompany our structural analysis, of which only half overlap with previously reported ones, we have now removed the Jensen data from the supplemental information, and reference the paper directly instead.

	Jensen BJP 2012			our work				
	EC50 (nM)	EC50 fold WT	% WT expression	IC50 nM	IC50 fold WT	I _{max}	I _{max} fold WT	% WT expression
WT	0.7	1.0	100	0.4	1.0	100.0	1.0	100
Q91A	1.4	2.0	53	0.1	0.5	77.0	2.3	110
H283A	1.2	1.8	60	2.8	4.7	45.4	1.7	106
Y113A	13.0	19.0	21	N/A		N/A		65
Y114A	0.9	1.3	17	2.5	6.0	74.5	2.7	79
Y172A	8.5	12.0	9	1.3	5.3	82.3	2.6	77
W251A	6.6	9.4	15	N/A		N/A		81
F254A	0.9	1.2	27	1.7	8.2	46.8	1.4	104
E286A	2.2	3.2	27	26.1	61.8	74.4	2.7	130
Y184A	14.0	20.0	94	13.6	54.3	42.7	1.3	99

Table R2. Comparison of functional characteristics for CCR8 mutants reported both in Jensen BJP 2012 and tested in this study. Values in each column are colored using a green-yellow-red color scale to reflect the degree of divergence from wild-type properties within each dataset.

2. The authors did a 200-ns molecular dynamics simulation to learn the conformational dynamics of CCL1 and CCR8 ECL3 in the CCL1-CCR8 complex. But 200 ns may be too short to confirm the interaction between the chemokine and the receptor. It will be more robust for a time scale longer than 1 μ s in the MD simulation.

We agree with the Reviewer that two classical Molecular Dynamics simulations of 250 ns may be too short to confirm the CCL1-CCR8 interactions. At the request of the Reviewer, we thus completed an additional four Gaussian-accelerated Molecular Dynamics (GaMD) simulations totaling 2.8 μ s of simulation, thereby bringing the overall total simulation time to 3 μ s. We believe that the increased simulation time in addition to the use of GaMD, an unbiased enhanced sampling MD method, provides sufficient data to further confirm that the engineered disulfide between CCL1 and CCR8 does not substantially alter the structure of the CCL1-CCR8 complex or the interactions between the chemokine and the receptor.

The longer simulations demonstrate the same behavior as our original simulations, though show additional conformational heterogeneity, as expected from longer simulations and the use of enhanced sampling permitted by GaMD. We have now updated **Supplementary Figure 9** to show the longer GaMD simulations and modified the figure legends and manuscript text accordingly to describe the GaMD method and provide an updated description of our results, see lines 197-202.

3. As the authors have performed MD simulation of CCL1 interacting with CCR8, MD simulation of Fab1 bound to CCR8 should be performed. So that the conformation dynamics underlying the mechanism of CCR8 activation can be elucidated more deeply.

We agree with the reviewer that MD simulations could be used to more deeply understand the conformational dynamics underlying CCL1-induced activation of CCR8 and how binding of Fab1 may affect this, but we believe this would require significantly more simulation work and experimental validation that are beyond the scope of this publication.

Our main intent with the molecular dynamics simulations was to validate that the engineered disulfide between CCL1 and CCR8 did not substantially alter the structure of the CCL1-CCR8 complex or the interactions between the chemokine and the receptor, as well as to assess the overall flexibility and dynamics of the globular CCL1 core and the persistence of the interactions between the N-terminus of CCL1 and the orthosteric pocket of CCR8. We did not intend to use the molecular dynamics simulations to interpret the conformational dynamics underlying receptor activation and how binding of the Fab1 may affect this.

We believe that part of the confusion as to our intended use of MD simulations stems from the fact that we explicitly mentioned observed differences in CCR8 ECL3 conformation and hydrogen-bonding interactions between CCR8 H267^{6,64} and CCL1 S58 in the paragraph preceding the CCL1-induced activation mechanism. While we think this singular finding is interesting, we agree it is somewhat anecdotal and disconnected from the rest of the manuscript. As such, we have now removed the MD description of the CCR8 ECL3 H267^{6,64} - CCL1 S58 interaction from the manuscript and now explicitly state the rationale for running MD simulations and the resulting outcome and conclusions, see lines 197-202.

Minor points:

1. The mass of G α is larger than G β . But the position of G β is higher than G α as shown in supplementary Fig 4. The authors should check again the labels in the SDS-PAGE gel image. In addition, the band of scFv16 has been shown in the SDS-PAGE gel image, so the peak of scFv16 also should be displayed in the SEC elution profile.

We thank the Reviewer for pointing out this error in figure labeling and for asking us to expand on important technical aspects. We have now expanded the SEC profile in **Supplementary Figure 4a** to include the scFv16 elution and updated the labels in **Supplementary Figure 4b**.

2. In supplementary Fig 5b, the cryo-EM density map of $\alpha 5$ helix of Gi protein should be shown.

We agree that this should be included and thank the Reviewer for pointing this out. We have now added a new panel **e** to **Supplementary Figure 5** (shown below) to show the cryo-EM density map of the $\alpha 5$ helix of Gi.

$\alpha 5$ helix of G_i

Figure R3 (now included as **Supplementary Fig. 5e**) *Cryo-EM density map of the alpha 5 helix of Galpha i.*

3. In supplementary Fig 10c, the CCR8-Gi interface is compared with other receptors of CCR subfamily. The residues presented in sticks should be labeled.

We thank the Reviewer for pointing this out and have updated **Supplementary Figure 10c** accordingly.

4. In supplementary Fig 7b, the structure of human 5-hydroxytryptamine 2B (5-HT2B) was used for comparison. Could the authors explain the reason for choosing 5-HT2B in the comparison?

We compared all representative structures in the Protein Data Bank of class A GPCRs in complex with Fabs engaging the receptor extracellular loops (i.e. HT2B-Fab PDB ID 5TUD; AT2R-Fab, PDB ID 5XJM; EP4R-Fab, PDB ID 5YWY; S1PR3-Fab, PDB ID 7C4S; GPR20-Fab46, PDB ID 8HS2) to our Fab1-CCR8 structure. We noticed that the HT2B - Fab P2C2 complex structure stood out as being the only one where the Fab also engages the ECL2 β -hairpin through antiparallel β -strand pairing with its CDRH3, which is why we chose to highlight it in our manuscript. To provide this additional context to the readers, we have now updated this section of the manuscript (see lines 148-151) and include a comparison to all representative structures of class A GPCRs in complex with Fabs engaging the receptor extracellular loops, as shown below and now included in **Supplementary Fig 7b**.

Figure R4 (now included in the updated **Supplementary Fig. 7b**) **Comparison of representative structures of class A GPCRs in complex with antibodies targeting the receptor extracellular loops.** Top row, left to right: Fab1-CCR8 (this work), HT2B-Fab (PDB ID: 5TUD); AT2R-Fab (PDB ID: 5XJM). Bottom row, left to right: EP4R-Fab (PDB ID: 5YWY), S1PR3-Fab (PDB ID: 7C4S), GPR20-Fab46 (PDB ID: 8HS2).

5. The references (36, 40, 41) of previously defined chemokine recognition sites mentioned in line 161 should be revised. The authors can either refer to the CCR8 homologue such as CCR1, CCR5, and CCR6, or cite review articles summarizing the recognition modes of chemokine receptors.

The original references 36, 40 and 41 referred to Qin et al. Science 2014, Scholten et al. Brit J Pharmacol 2012 and Kufareva Annu Rev Biophys 2017. We chose to cite Qin et al. as it described one of the first chemokine ligand - receptor complexes and because CXCR4 was the very first inactive chemokine receptor structure to be solved (Wu et al. Science 2010). We cited Scholten et al. because in this paper the CRS nomenclature was first introduced ("We propose to use the term chemokine recognition site 1 (CRS1), instead of site I often used in the literature, to avoid confusion with binding sites in the transmembrane (TM) pockets for small molecules."). Finally, we cited Kufareva et al. because this review article does summarize chemokine recognition sites ("In this model, the receptor N-terminus binds to the chemokine core (an interaction often referred to as chemokine recognition site 1, or CRS1) while the chemokine N terminus binds in the pocket of the receptor TM helical domain (chemokine recognition site 2, or CRS2).").

At the request of the Reviewer, we have now removed the Qin et al. and Scholten et al. references and now include Isaikina et al. Science Advances 2021, Zhang et al. Nat Comm 2021 and Shao et al. Nat Chem Bio 2022, which describe chemokine ligand-receptor complex structures for CCR5 or CCR1.

6. In line 176-177, the authors stated that a previously unobserved feature in CCR8 is the CCL1 N-terminus folds back up away from the receptor core towards the extracellular regions. I am not quite sure about this conformation since the coordinates of CCL1-CCR8-Gi is not shown in the revision materials. But as shown in supplementary Fig 8, the pose of CCL20 in its N-terminus bound to CCR6 is similar to that of CCL1 in CCR8, showing a “hook-like” conformation. The author may need to compare the binding pose of CCL1 and CCL20 in more details.

We thank the Reviewer for bringing this up and for their suggestion to further compare the binding pose of CCL1 and CCL20. We have now included an additional panel **g** in **Supplementary Figure 8**, shown below, to reveal in more detail the distinct conformation of CCL1 relative to other CC chemokine ligands. We hope the Reviewer and readers can now better appreciate that CCL1 appears to be the only chemokine for which the N-terminus folds back upwards, away from the receptor core and towards the extracellular space, thereby forming a "hook-like" conformation that is distinct from CCL20 bound to CCR6 and other published C-C chemokine ligand-receptor structures.

g

Figure R5 (now included as **Supplementary Fig. 8g**) **Close-up comparison of CC chemokine ligand binding poses in the receptor orthosteric pocket.** Compared to other published structures, CCL1 displays a unique "hook-like" conformation where its N-terminus folds up, away from the receptor core.

7. In line 213-214, the description of interactions directly and indirectly rearranges Y1133.32, Y1143.33, and F1173.36 side chains is not clear. It will be more readable to clarify which residues form direct interaction and which residues mediate indirect interaction.

We agree with the Reviewer that this merits clarification, as only Y114^{3.33} directly interacts with CCL1. We have now rewritten this part of the results section to provide a clear description of the rearrangements we observe between our inactive and active structures and, distinguishing between direct and indirect interactions, and moved a revised description of the putative activation mechanism to the discussion section, see lines 212-216 and 280-303.

8. In the study of epitope mapping of mAb1 by flow cytometry, why did the authors select CCR5 for the design of constructs encoding for human CCR8-CCR5 chimeras?

We wanted to ensure that our chimeric constructs would fold properly and express well. We compared the sequences of all human CCR family members and found that taken together, CCR4 and CCR5 had the highest and second-highest sequence identity and similarity to CCR8, both for the full receptor sequence and in particular for the extracellular loop sequences, respectively (see **Figure R6** below). Phylogenetic tree clustering of the human CCR sequences indicated that CCR4 clustered the closest to CCR8, while CCR5 clustered further away (see **Figure R7** below). We chose CCR5 to maximize the chance of success of our chimeric designs while providing sufficient sequence divergence to properly assess the specificity of our antibody. We now mention the rationale for choosing CCR5 to design CCR8 chimeras in the manuscript, see lines 93-94.

Full sequence similarity / identity matrix

Similarities are on the lower-left side of the table, and identities on the upper-right.

	1	2	3	4	5	6	7	8	9	10
1. [Human] CCR1	-	43	62	43	51	29	32	38	28	25
2. [Human] CCR2	57	-	41	40	60	28	27	32	25	22
3. [Human] CCR3	77	55	-	41	47	29	29	35	28	23
4. [Human] CCR4	60	57	58	-	44	33	31	41	30	24
5. [Human] CCR5	67	67	65	61	-	28	29	38	29	24
6. [Human] CCR6	47	45	47	51	47	-	34	28	31	27
7. [Human] CCR7	47	44	46	48	47	49	-	25	33	31
8. [Human] CCR8	58	47	55	61	55	47	45	-	26	21
9. [Human] CCR9	43	44	46	47	48	49	52	45	-	26
10. [Human] CCR10	41	38	39	41	40	45	46	38	41	-

Extracellular loop sequence similarity / identity matrix

Similarities are on the lower-left side of the table, and identities on the upper-right.

	1	2	3	4	5	6	7	8	9	10
1. [Human] CCR1	-	27	33	25	46	19	33	25	14	11
2. [Human] CCR2	42	-	15	28	35	21	25	32	14	7
3. [Human] CCR3	58	31	-	21	25	15	22	17	14	7
4. [Human] CCR4	25	36	21	-	29	23	15	35	21	7
5. [Human] CCR5	67	42	46	33	-	12	19	33	18	7
6. [Human] CCR6	23	25	27	35	15	-	17	23	21	18
7. [Human] CCR7	37	32	41	22	30	24	-	15	20	10
8. [Human] CCR8	38	40	33	48	42	35	22	-	25	18
9. [Human] CCR9	29	34	32	21	39	25	33	36	-	18
10. [Human] CCR10	18	17	18	18	14	36	20	25	36	-

Figure R6 Sequence similarity and identity matrices for human CC chemokine receptors. The full receptor sequences (top panel) or only the extracellular loop sequences (bottom panel) were used to calculate identities and similarities. Tables were generated using the www.gpcrdb.com sequence analysis tool.

Phylogenetic tree

Figure R7 Phylogenetic tree of all human CCR receptors. The CCR5 and CCR8 receptors are boxed in red and blue, respectively. Generated using the www.gpcrdb.com sequence analysis tool.

9. In the legend of Fig 1d, “CCL1-CCR8-Gi1” should be changed to “CCL1-CCR8-Gi1-scFv16”.

We have now corrected this.

10. In the legend of Fig 5c, what is the meaning of xx circles (line 846)?

We apologize for this oversight during the final editing of our figures and figure legends. We meant "green" and have now corrected this.

Reviewer #2 (Remarks to the Author):

In their manuscript Sun et al describe the detailed structure of chemokine receptor CCR8 conjugated to its ligand CCL1. In addition, they describe the production of a specific antibody to CCR8 that inhibits CCL1-induced calcium signaling through CCR8. The antibody does not bind to the N-terminus of the receptor, like many other described anti chemokine receptor antibodies, but has a long CDRH3 that allows it to interact with the extracellular loops (mainly ECL2). This might be compared to nanobodies that have the advantage to bind better to epitopes that are oriented more in depth in their target proteins. Interestingly, the authors describe a unique orientation of the N-terminus of CCL1 when bound to CCR8 compared to other chemokine ligand receptor pairs. This unique orientation of the N-terminal residue creates a chemokine recognition site on the receptor (CRS2.5) which is so far unique in the chemokine – chemokine receptor interactions.

We thank the Reviewer for their appreciation of the detail of our work and its relevance and novelty.

Main comments:

1. A unique biological characteristic of the CCL1 – CCR8 pair is that CCL1, in addition to its typical chemokine activities such as chemotaxis and induction of calcium signals, also has anti-apoptotic activity on T cells. This is in particular important in the context of the role of Treg cells in the tumor environment. Authors show that calcium signaling through CCR8 is inhibited by their antibody even though the antibody is still able to bind to a receptor-ligand pair. Is also the anti-apoptotic activity of CCL1 inhibited by the antibody. This would be highly interesting with regards to an application potential of the antibody.

The anti-apoptotic function of CCL1 has been demonstrated in the literature in the context of dexamethasone treatment of BW5147 cells, a mouse T lymphocyte line 1 (Snick et al. *J. Immunol.* 1996; Spinetti et al. *Journal of leukocyte biology* 2003; Louahed et al. *Eur. J. Immunol.* 2003; Denis, C. et al. *Plos One* 2012; Van Damme et al. *J Immunother Cancer* 2021). However, our anti-human CCR8 antibody mAb1 is not cross-reactive with murine CCR8, preventing us from using this system. We instead tested the ability of dexamethasone to induce apoptosis in CCR8-expressing human Jurkat T cells, but did not observe any apoptosis in this setting (**Figure R8**).

Figure R8 Dexamethasone does not induce apoptosis in Jurkat.hCCR8 cells. Jurkat.hCCR8 cells were treated with dexamethasone at the indicated concentration for 24h. Induction of apoptosis was evaluated by annexin V staining and flow cytometry analysis.

In an attempt to further address the reviewer's question, we utilized an anti-Fas antibody, which did induce apoptosis in Jurkat cells, and attempted to inhibit Fas-induced apoptosis by stimulating the cells

with CCL1. However, we did not observe any protection from apoptosis by CCL1/CCR8, preventing us from testing an inhibitory effect of the anti-CCR8 mAb1 (**Figure R9**). As such, we did not include this data in the revised manuscript.

Figure R9 CCL1 does not block anti-Fas antibody induced apoptosis in Jurkat.hCCR8 cells. (A)

Jurkat.hCCR8 cells were treated with the apoptosis-inducing anti-Fas antibody (clone CH11) at the indicated concentration for 24h. Induction of apoptosis was evaluated by annexin V staining and flow cytometry analysis. (B) Jurkat.hCCR8 cells were pre-incubated with CCL1 for 10 min at the indicated concentration and then treated with anti-Fas antibody (1ng/ml) for 24h. Cell apoptosis was evaluated by annexin V staining and flow cytometry analysis.

Louahed *et al. Eur. J. Immunol.* 2003 and Spinetti *et al. Journal of leukocyte biology* 2003 demonstrated that the anti-apoptotic effect of CCL1/CCR8 is ERK-dependent and that CCL1 induced ERK1/2 phosphorylation. We therefore assessed ERK phosphorylation upon CCL1 treatment in CCR8-expressing Jurkat cells and found robust p-ERK induction within 5 min of stimulation. We next pre-incubated hCCR8.Jurkat cells with increasing concentrations of CCR8 mAb1 before stimulating the cells with 20 nM CCL1. In this assay we observed a dose-dependent inhibition of ERK phosphorylation, and could thus confirm that mAb1 blocked CCL1-mediated activation of the MAPK pathway in addition to blocking Calcium signaling (**Figure R10**), and we have now included this data in the manuscript in **Figure 1c** and **Supplementary Figure 2e**.

Figure R10 (now included as Figure 1c and Supplementary Figure 2e) mAb1 inhibits ERK phosphorylation by CCL1 in huCCR8.Jurkat cells. huCCR8.Jurkat cells were incubated with anti-CCR8 antibody mAb1 for 30min, then 20nM CCL1 was added for 5min. Cell lysates were analyzed by western blotting using anti-phospho-ERK or anti-ERK antibody (left panel). Bands were quantified using the image studio lite software and pERK values were normalized to the corresponding total ERK for each condition (right panel).

2. There is discussion in the field on the interaction between CCL18 and CCR8. Can the authors conclude anything from their structural analyses on the potential of CCL18 to bind (or not to bind or activate) CCR8?

To further investigate whether CCL18 can signal through CCR8, we tested whether CCL18 could induce a Ca^{2+} flux in our CHO.hCCR8 stable cell line using the FLIPR assay and whether it could induce cAMP inhibition in HEK293 cells transiently transfected with WT hCCR8. While CCL1 induced a dose-

dependent response in both assays as expected, we did not observe any response for CCL18 (**Figure R11**).

Figure R11 (now included as **Supplementary Fig. 14**) **Functional assessment of CCL18 signaling through CCR8.** Ca²⁺ flux measurements in a CHO.hCCR8 stable cell line using the FLIPR assay (left panel) and cAMP inhibition measurements in HEK293 cells transiently transfected with WT CCR8 (right panel) were performed to assess whether CCL18 can elicit a CCR8-dependent response.

Of note, in Islam et al J Exp Med 2013, the authors demonstrated CCL1- and CCL18-induced migration of 4DE4 mouse pre-B cell lines transfected with human CCR8, with a peak migration at 10nM for both ligands, but a 5-fold weaker response for CCL18. The authors also showed that in competitive radiolabeled chemokine binding experiments, the relative binding affinity of CCL18 was about 10-fold lower than CCL1. However, in Liu et al Biochemical Pharmacology 2021, using competitive fluorescently labeled chemokine binding experiments, the authors were unable to show any binding of CCL18 up to 1 µg/mL in Jurkat cells stably expressing high levels of CCR8.

At the Reviewer's request, we further investigated the potential of CCL18 to bind from a structural perspective by generating models for CCL1, CCL18 or MC148 (a CCR8-specific viral chemokine) in complex with CCR8 using *AlphaFold2 multimer*, see **Figure R12**. Strikingly, the predicted AlphaFold model of the CCL1-CCR8 complex matches our experimental structure well, with the chemokine adopting a very similar binding pose, with its N-terminus in a hook-like conformation, similar to what we observe. However, unlike the AlphaFold prediction for CCL1-CCR8, the AlphaFold prediction for CCL18-CCR8 suggests an aberrant binding pose for the chemokine, with the N-terminus not engaging the receptor core and the globular core of the chemokine in a relative orientation that does not correspond to the typical chemokine binding pose. The MC148-CCR8 AlphaFold model suggests a more reasonable binding pose for this viral chemokine, but which is still quite different from how CCL1 engages CCR8. As such it is difficult to assess the validity of these models, other than making the obvious conclusion that the much shorter N-terminus of MC148 cannot penetrate the receptor binding pocket regardless of the binding pose and hence would not activate the receptor, in line with our structural interpretations from the experimental structure.

Figure R12 Structural comparison of experimentally determined and computationally predicted chemokine-receptor complexes. Left to right: CCL1-CCR8 cryoEM structure (this work), CCL1-CCR8 AlphaFold model, CCL18-CCR8 AlphaFold model, MC148-CCR8 AlphaFold model.

We also generated an additional CCL18-CCR8 model by first generating an AlphaFold model of CCL18 alone and then docking it in our CCL1-CCR8 structure with CCL1 removed (**Figure R13**). We compared this model to our CCL1-CCR8 structure to further assess the potential of CCL18 to bind CCR8. Both CCL1 and CCL18 have an N-terminal sequence preceding the CC motif of equal length, but no matching residues in this region. Inspection of this region suggests a reasonable binding pose for CCL18 but does not point to any obvious potential interactions that would activate the receptor in a similar way as CCL1.

Figure R13 Comparison of experimentally determined CCL1 binding pose and docked CCL18 binding pose in the CCL1-CCR8 cryoEM structure. The CCL1-CCR8 cryoEM structure (this work) is shown in blue and teal/orange, while the AlphaFold CCL1 model docked into the CCR8 structure is shown in pink and green.

Taken together, we feel that the evidence is not overwhelmingly strong to support CCL18 being a genuine ligand for CCR8, but we readily acknowledge that none of the structural interpretations above provides conclusive or definitive answers, and we believe only our functional data provides experimental evidence worth including in the manuscript. Based on this, we now include the functional data testing CCL18 signaling in our manuscript in a new **Supplementary Figure 14**, and have accordingly revised the discussion section about CCL18 in our manuscript, see lines 305-311.

Minor comments:

1. Suppl. Fig. 7a: check the ligand receptor pair for CCR5 in the figure and for CCR2 in the legend. They are probably wrong.

We thank the Reviewer for catching this mistake. We have now corrected the labels in **Supplementary Figure 7a**.

2. Line 280: are these losses in potency correct? I don't find them in the Suppl. Table.

We thank the Reviewer for catching this mistake. We meant to state that alanine mutation of Y113^{3,32}, F290^{7,43} and W251^{6,48} resulted in respective 19-, 6.4- and 9.4-fold losses in potency. However, since we have now generated our own mutagenesis data (see **Figures 3e,f and 4f,g**), we have updated this section of the manuscript to discuss our data and reference Jensen et al BJP 2012 directly instead, see lines 296-303.

3. Correction of some typo's: Line 204: CCL1's instead of CCL's; line 541: delete "between"; Suppl Table 4: correct PMBC to PBMC

We thank the reviewer for catching these typos and have now corrected them.

Reviewer #3 (Remarks to the Author):

The manuscript by Sun et al describes structure/function studies of CCR8. Two experimental structures are determined: 1) CCR8 in complex with Gi and agonist ligand CCL1 (which is engineered to bind CCR8 covalently via disulfide bridge); and 2) CCR8 in complex with antagonist antibody Fab1. By comparing these two structures and performing real-time binding assay the authors conclude that CCL1 binding to CCR8 follows a two-step mechanism, whereby low affinity binding at CRS1 is followed by high affinity binding at CRS2 that requires conformational changes in the receptor.

We thank the Reviewer for their evaluation of our work.

The manuscript also reports on computational MD simulations of CCL1-CCR8-Gi complex with and without the engineered disulfide bond between the ligand and the receptor. For each construct two replicates were run for relatively short, 250ns time. The goal of these simulations was to confirm that the conformational dynamics of CCL1 was not affected by the disulfide. The observation from the simulations was that removal of the engineered disulfide allowed the CCR8 N-terminus to move away from CCL1, but did not alter the conformational heterogeneity of CCL1, suggesting that the engineered disulfide does not affect how CCL1 engages the receptor core.

1. First, this study and conclusion seems to be somewhat disconnected from the rest of the manuscript. How does this inference add to the experiments that are reported? Perhaps, some mechanistic predictions could have been made from these simulations and tested experimentally?

Our main intention with the MD simulations was to validate that the engineered disulfide between CCL1 and CCR8 did not substantially alter the structure of the CCL1-CCR8 complex or the persistence of the interactions between the N-terminus of CCL1 and the orthosteric pocket of CCR8, while also probing the overall flexibility and dynamics of the globular CCL1 core to rationalize the lower resolution we see for that region of the chemokine. We did not intend to use MD simulations to infer mechanistic predictions, as we believe this would require significantly more simulation work and experimental validation that are beyond the scope of this publication.

2. Secondly, it seems to me that to make such strong conclusions, one needs to simulate the system for much longer time than 250ns. In general, the authors should make a better effort to present the significance of the presented simulations.

We agree with the Reviewer that classical Molecular Dynamics simulations of 250 ns may be too short to confirm the CCL1-CCR8 interactions. We have now completed four Gaussian-accelerated Molecular Dynamics (GaMD) simulations totaling 2.8 μ s of simulation, thereby bringing the overall total simulation time to 3 μ s. We believe that the increased simulation time, in addition to the use of GaMD, an unbiased enhanced sampling MD method, provides sufficient data to confirm that the engineered disulfide between CCL1 and CCR8 does not substantially alter the structure of the CCL1-CCR8 complex or the interactions between the chemokine and the receptor. The longer simulations demonstrate the same behavior as our original simulations, though show additional conformational heterogeneity, as expected from longer simulations and the use of enhanced sampling permitted by GaMD. We have now updated **Supplementary Figure 9** to show the longer GaMD simulations.

We agree with the Reviewer that the use and significance of the simulations could be presented better. As stated earlier, our main intent with the MD simulations was to validate that the engineered disulfide between CCL1 and CCR8 did not substantially alter the structure of the CCL1-CCR8 complex or the interactions between the chemokine and the receptor, as well as to assess the overall flexibility and dynamics of the globular CCL1 core and the persistence of the interactions between the N-terminus of CCL1 and the orthosteric pocket of CCR8.

We believe that part of the confusion and disconnect as to our intended use of MD simulations stems from the fact that we explicitly mentioned observed differences in CCR8 ECL3 conformation and hydrogen-bonding interactions between CCR8 H267^{6.64} and CCL1 S58 in the paragraph preceding the CCL1-induced activation mechanism. While we think this singular finding is interesting, we agree it is somewhat anecdotal and disconnected from the rest of the manuscript. As such, we have now removed the MD description of the CCR8 ECL3 H267^{6.64} - CCL1 S58 interaction from the manuscript and now explicitly state the rationale for running MD simulations and the resulting outcome and conclusions, see lines 197-202.

3. Overall, lot of structural comparisons are made, based on the cryo-EM models. The authors should be careful that these are just isolated structures and therefore inferring on some dynamic mechanisms from such comparisons could be misleading. For example, there is a sentence: "Careful analysis of the receptor-chemokine interactions allows us to rationalize how signal transmission from CRS2 to the canonical GPCR microswitches occurs to enable the outward motion of the intracellular side of TM6, the hallmark of GPCR activation." Seems like that the authors are describing a very complicated allosteric mechanism from simple structural comparison. I would advise against this. It is best just to use simple comparative description stating what is different between the two structures.

The canonical microswitches and associated activation mechanisms of class A GPCRs have been extensively studied and are often described in publications comparing active and inactive class A GPCR structures in general and inactive versus active CC chemokine receptor structures in particular (e.g. Liu et al Nature 2020, Isaikina et al Science Advances 2021, Shao et al Cell Discovery 2022). However, we understand the Reviewers' concern that our description of the results inferring dynamic and allosteric mechanisms from static structures could be misleading. Accordingly, we are now describing the observed changes between our two structures in the Results section (see lines 190-195, 210-216, 218-224 and 232-244) and provide a revised description of the putative activation mechanism in the Discussion section (see lines 280-303).

REVIEWERS' COMMENTS

Reviewer #1 (Remarks to the Author):

The authors have carefully conducted experiments and included additional data to address my concerns. In particular, they conducted their own functional assays using mutants that affected ligand binding and mutants that affected GPCR signaling. As a result, the added information makes the manuscript complete and well rounded. Several questions were also answered or the issues clarified. About the only requested experiment that was not conducted is the MD simulations of the Fab1 bound receptor. For this the authors provided an explanation which seems reasonable. I found the revised manuscript improved, and I have no additional questions.

Reviewer #2 (Remarks to the Author):

In their revised paper the authors carefully and adequately addressed my concerns and questions.

Reviewer #3 (Remarks to the Author):

The revised manuscript has addressed all my previous comments.

Response to Reviewers

We thank all our Reviewers for their support of this work and for their thoughtful evaluation and constructive feedback. The Reviewers' feedback has been very valuable to help us further elevate the presentation and impact of our work and we are glad to see that the Reviewers are fully satisfied with our revisions and responses.

REVIEWERS' COMMENTS

Reviewer #1 (Remarks to the Author):

The authors have carefully conducted experiments and included additional data to address my concerns. In particular, they conducted their own functional assays using mutants that affected ligand binding and mutants that affected GPCR signaling. As a result, the added information makes the manuscript complete and well rounded. Several questions were also answered or the issues clarified. About the only requested experiment that was not conducted is the MD simulations of the Fab1 bound receptor. For this the authors provided an explanation which seems reasonable. I found the revised manuscript improved, and I have no additional questions.

We thank the Reviewer for their evaluation of our revised work. We are glad the Reviewer is fully satisfied with the answers we provided to their questions and with our revised manuscript.

Reviewer #2 (Remarks to the Author):

In their revised paper the authors carefully and adequately addressed my concerns and questions.

We thank the Reviewer for their evaluation of our revised work. We are glad the Reviewer is fully satisfied with the answers we provided to their questions and with our revised manuscript.

Reviewer #3 (Remarks to the Author):

The revised manuscript has addressed all my previous comments.

We thank the Reviewer for their evaluation of our revised work. We are glad the Reviewer is fully satisfied with the answers we provided to their questions and with our revised manuscript.